# Iron-mediated post-transcriptional regulation in *Toxoplasma gondii*

**Megan A. Sloan**\*, **Adam Scott, Dana Aghabi, Lucia Mrvova, Clare R. Harding**⬛\*

School of Infection and Immunity, University of Glasgow, Glasgow, United Kingdom

\* megan.sloan@glasgow.ac.uk (MAS); clare.harding@glasgow.ac.uk (CRH)

## Abstract

Iron is required to support almost all life; however, levels must be carefully regulated to maintain homeostasis. Although the obligate parasite *Toxoplasma gondii* requires iron, how it responds upon iron limitation has not been investigated. Here, we show that iron depletion triggers significant transcriptional changes in the parasite, including in iron-dependent pathways. We find that a subset of *T. gondii* transcripts contain stem-loop structures, which have been associated with post-transcriptional iron-mediated regulation in other cellular systems. We validate one of these (found in the 3' UTR of TGME49_261720) using a reporter cell line. We show that the presence of the stem-loop-containing UTR is sufficient to confer accumulation at the transcript and protein levels under low iron. This response is dose and time-dependent and is specific for iron. The accumulation of transcript is likely driven by an increased reporter mRNA stability under low iron. Interestingly, we find iron-mediated changes in mRNA stability in around 400 genes. To examine the potential mechanism of this stability, we tested aconitase interaction with mRNA in low iron and found 43 enriched transcripts, but no specific interaction with our reporter UTR. However, the endogenous UTR led to maintenance of protein levels and increased survival of the parasite under low iron. Our data demonstrate the existence of iron-mediated post-transcriptional regulation in *Toxoplasma* for the first time; and suggests iron-mediated regulation may be important to the parasite in low iron environments.

## Author summary

Although iron is essential for most cells, excess can be toxic, so uptake and storage must be carefully controlled. The ability to respond to iron is particularly important for parasites, such as *Toxoplasma gondii,* which must replicate within host cells where iron levels can fluctuate. Here, we have investigated how the parasite responds to a lack of available iron. We find that while gene expression changes significantly, this does not appear to give us the whole picture. By examining predicted structures in the messages that code for proteins, we find that many *T. gondii* genes contain a small stem-loop structure, which has been shown to be important in iron-responsiveness in other organisms. We show that this structure helps the parasite to accumulate specific proteins, which could potentially help the cells survive in low iron. By investigating this system in *T. gondii*, we gain a better understanding of how the parasite senses and responds to its environment.

**Data availability statement:** The data that support the findings of this study are publicly available from EBI ENA with the identifier(s) PRJEB67890, PRJEB83011 and PRJEB83013. All other data are within the manuscript and its Supporting Information files.

**Funding:** M.A.S and L.M are funded by an Early Career Award from the Wellcome Trust (225677/Z/22/Z). D.A. is funded by the Wellcome IIB PhD program (218518/Z/19/Z). C.R.H. is funded by a Sir Henry Dale Fellowship from the Wellcome Trust and the Royal Society (213455/Z/18/Z). The funders had no role in study design, data collection and analysis, decision to publish, or preparation of the manuscript.

**Competing interests:** The authors have declared that no competing interests exist.

## Introduction

*Toxoplasma* is a highly successful obligate intracellular parasite with a broad host range. Infection in humans is typically asymptomatic, although in pregnant or immunocompromised people, infection can have severe consequences. During infection, *T. gondii* invades and replicates within multiple cell types–exposing the parasite to rapidly changing levels of the key nutrients required to support replication. One of these essential nutrients is iron, *Toxoplasma gondii* requires iron to power cellular metabolism and replication [1,2]. We have found that *Toxoplasma* can detoxify excess iron by storing it within a membrane-bound compartment through the action of the vacuolar iron transporter (VIT), and that this storage is required to protect cells from excess iron [3,4]. However, the parasite is likely to be exposed to limiting iron levels during infection as some tissues have low levels of available iron, and mammalian hosts restrict iron availability during infection [5,6]. This means the parasite must sense and respond to changing levels of iron to maintain pathogenesis, however, the mechanisms underlying this remain unknown.

Due to the importance of iron across kingdoms of life, regulation of iron uptake, usage and storage is highly conserved [7–10]. However, mechanisms used to respond to iron stress are frequently distinct between organisms and range from control at the transcriptional [10], post-transcriptional and protein [11] levels. These pathways allow cells to regulate key proteins in response to low iron to promote survival.

One of the best studied mechanisms for iron-mediated regulation in mammalian cells is the post-transcriptional IRE/IRP system. Transcripts of genes involved in cellular iron homeostasis (such as the transferrin receptor, TfR, the main pathway of cellular iron uptake) contain short stem-loop regions, called iron-response elements (IREs) [12,13]. In low iron conditions, the metabolic enzyme aconitase (also called iron-regulatory protein 1, IRP1) loses its iron-sulphur (FeS) cluster cofactor. This abrogates its enzyme activity, and instead aconitase moonlights as an RNA-binding protein where it binds to IREs present in the untranslated regions (UTRs) of specific genes [14–18]. Crucially, the location of the IRE within the transcript determines whether the transcript is repressed or translation is promoted. The presence of IREs in the 3' UTR of genes acts to stabilise transcripts and promotes protein production [15,19]. Whereas binding in the 5' UTR, found in transcript of the iron storage protein ferritin, is thought to block access of the translation apparatus [14,20,21], inhibiting translation. This system allows cells to respond to iron stress by upregulating iron import (via TfR) and downregulating storage, leading to greater iron availability for cellular processes.

The aconitase/IRE system has been reported to regulate iron response in several organisms beyond mammals, including invertebrates and bacteria [22–24] however with limited supporting evidence. In plants, the role of aconitase in iron-response remains unclear; in *Arabidopsis thaliana* aconitases can bind RNA [25], however this did not appear to directly modulate ferritin abundance [26], possibly explained by functional redundancy between the three aconitase homologs. Iron-mediated regulation beyond model organisms is generally less well understood. In the kinetoplast *Trypanosoma brucei,* an mRNA binding protein binds and stabilises the transcript of the parasite transferrin receptor under low iron to promote iron uptake [27]. Similar systems have been proposed in the protozoan parasites *Trichomonas vaginalis* [28,29] and *Giardia duodenalis* [30]. Within the apicomplexan family, there is some evidence of a functional IRE system. In *Plasmodium*, the causative agent of malaria, IRE-like structures have been identified and shown to interact with parasite aconitase [31,32]. However, these experiments were performed *in vitro* using bacterially purified proteins, and the existence of a functional system in cells was not established.

Here we perform RNA-sequencing on iron-deprived parasites and find global transcriptional changes, including alterations in iron-dependent pathways. To identify possible mechanisms for parasites responses to low iron, we identify *in silico* IRE-like sequences in the *T. gondii* transcriptome. We chose to examine a predicted IRE from the 3' UTR of an unstudied gene, TGME49_261720, which is predicted to encode a zinc-iron permease, which we named ZFT (zinc and iron transporter). We show that this IRE-like containing UTR is required and sufficient to confer iron-responsivity to a reporter gene. We find that the *zft* UTR confers increased stability to mRNA under low iron and this increased stability is seen across 100s of transcripts. We test the ability of *T. gondii* aconitase to bind to this UTR, and while we see enrichment of predicted IREs in the bound transcripts, aconitase does not appear to be responsible for the iron-mediated regulation of *zft*. We further investigate the role of the *zft* UTR in the endogenous context; and find replacement of the UTR confers increased susceptibility to low iron. Here we present the first functional analysis of an IRE-like sequence in *T. gondii* and provide evidence for a role of post-transcriptional regulation in *T. gondii* iron homeostasis.

## Results

### The transcriptional response to iron deprivation

Transcriptional responses to iron deprivation are common across organisms [9,33,34]. To determine how iron deprivation impacts the transcriptome of *T. gondii,* we deprived parasites of iron by infecting primary human cells (HFF), pretreated for 24 h with the iron chelator deferoxamine (DFO). After 24 h, parasites were mechanically released, filtered to remove host debris and bulk RNA-sequencing performed (Fig 1A). We observed 1197 transcripts with higher abundance in parasites cultured in low iron conditions compared to untreated parasites, whilst just 44 were downregulated ($\log_2$ fold change $<-2$, adjusted *p*-value (padj) $< 0.05$) (S1 Table). This bias towards upregulation of transcripts was unexpected and was not seen upon oxidative stress induction [36] or upon downregulation of an essential metabolic regulator [37].

Gene ontology (GO) enrichment analyses were performed to gain an overview of processes potentially impacted by the observed transcriptomic changes (S1A Fig). The most significantly enriched GO terms associated with downregulated transcripts were GO:0006259 DNA metabolic processes (Bonferroni adjusted *p*-value 0.0001) and GO:0006261 DNA-dependent DNA replication (Bonferroni adjusted *p*-value 0.0004). This is in agreement with our previous work showing reduced parasite growth and proliferation upon DFO treatment [4] and likely reflects the iron requirements of multiple components of the DNA replication/repair machinery. Enriched GO terms associated with upregulated transcripts included terms associated with microtubule-associated transport. Looking closer at the genes involved, we found multiple dynamin and kinesins upregulated, potentially indicating a role in vesicular trafficking in iron response, however, the significance of this remains unclear.

We next examined if iron deprivation impacted transcript abundance of genes encoding predicted iron sulphur (FeS) cluster proteins, using the *in silico* prediction tool MetalPredator [38]. Of 60 FeS containing genes, we found 27 were significantly differentially expressed in our dataset (padj $< 0.05$) with 67% of these transcripts being downregulated (S2 Table), representing a significant ($p < 0.001$, Fisher's exact test) enrichment of predicted FeS-protein genes in the downregulated subset. *T. gondii* encodes three FeS biosynthesis pathways [2] of these, the impact of iron deprivation appears minimal on the cytosolic (CIA) and apicoplast (SUF) FeS synthesis pathways with 1/8 and 2/9 genes modestly regulated ($\log_2$ fold

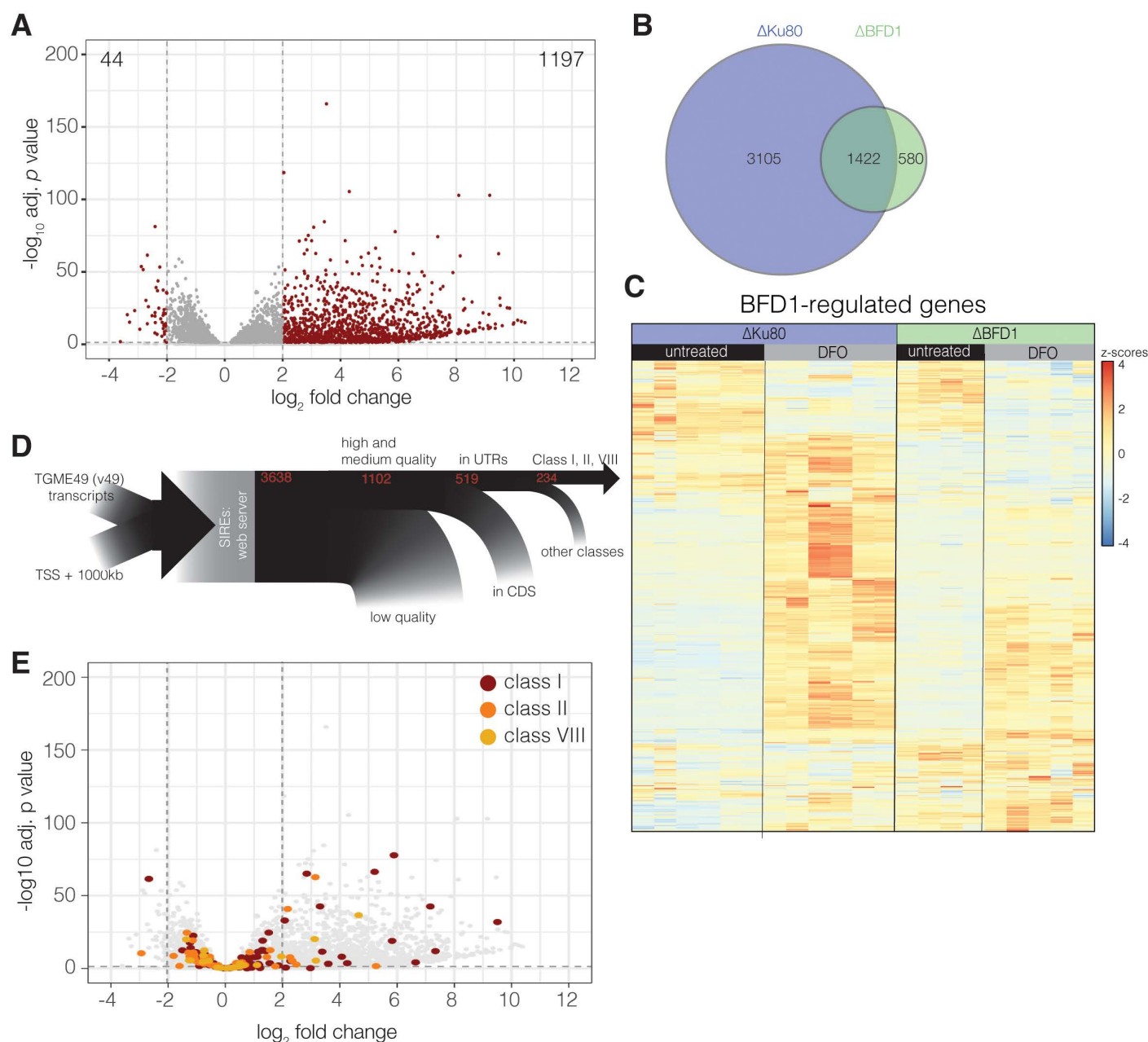

**Fig 1. The *T. gondii* transcriptional response to iron deprivation. A.** Volcano plot from RNAseq data comparing untreated ΔKu80 to parasites cultured in 100 μM DFO for 24 hours. Adjusted p-values from the Wald test with Benjamini and Hochberg correction. Cut-offs shown with dashed lines are p-adj < 0.05 and log$_2$ fold change of >2 or <−2. **B.** Venn diagram showing overlap between significantly regulated genes from DFO-treated parental and ΔBFD1 parasites. **C**. Heatmap from RNAseq dataset depicting z-scores of genes shown to be regulated by BFD1 (Waldman *et al.* 2020) [35], in parental and ΔBFD1, DFO treated and untreated parasites. **D.** Schematic showing the filtering process to identity transcripts containing IRE-like sequences in the *T. gondii* transcriptome. **E.** Volcano plot as in (**A**) where transcripts with IRE-like sequences in classes I (red), II (orange) and VIII (gold) are highlighted. See text for more details.

changes < 1, padj < 0.05, S3 Table). The largest effect was seen in the mitochondrial FeS synthesis pathway, with 8/12 genes in the ISC pathway differentially regulated including downregulation of the FeS scaffold ISU1 (TGME49_237560) and the proposed FeS carrier NFU4 (TGME49_212930). The FeS co-chaperone MGE1 (TGME49_265220) and ferredoxin

MFdx1 (TGME49_236000) were also downregulated, though fold changes were modest. However, ISD11 and NFS1, which provide sulphur to clusters, were upregulated. Supporting this, many of the downregulated FeS-containing genes are predicted to be mitochondrially localised [39] (S2 Table).

Of the upregulated transcripts, we saw bradyzoite stage specific genes including: BAG1 (log$_2$ fold change = 3.2, padj = 1.94E-10), ENO1 (log$_2$ fold change = 5.1, padj = 6.3E-18) and LDH2 (log$_2$ fold change = 6.4, padj = 6.87E-29). We also observed that the majority of transcripts found to be upregulated in bradyzoites in previous work [35] were also upregulated in iron-deprived parasites (S4 Table). The transcription factor BFD1 has been shown to accumulate during cell stress and was required and sufficient to drive parasite differentiation [35]. To identify iron regulated genes not related to differentiation, we repeated our RNA-sequencing experiment in a BFD1 knockout parasite line (S1B Fig). We observed 2002 genes with differential expression upon DFO treatment (padj < 0.05, S5 Table and S1B Fig), of these 1422 were also differentially expressed in RHΔKu80 upon DFO treatment (Fig 1C). Comparison of the log$_2$ fold changes of these genes demonstrates clear correlation between the two datasets (linear regression R$^2$ = 0.86, *p* < 0.0001, S1B Fig). We observed expression patterns in genes reported to have BFD1 binding sites are similar between experiments, with DFO-associated changes appearing 'dampened' in the BFD1 line (Fig 1D). Surprisingly, there were 256 genes with BFD1 binding sites which were upregulated in the BFD1 knockout line upon DFO treatment, suggesting these genes can also be regulated by other factors. GO/Biochemical pathway enrichment analysis of these genes did not highlight significantly enriched processes.

This provides further evidence that iron deprivation, similar to other nutrient deficiencies [40–43], can promote *Toxoplasma* differentiation. Given the strong, apparently non-specific, effects of iron deprivation on the *T. gondii* transcriptome, we decided to search for potential downstream post-transcriptional regulation which could mediate specific responses to low iron.

## The *T. gondii* transcriptome contains IRE-like sequences

We identified transcripts containing putative iron response elements in the *T. gondii* using the online RNA structural prediction tool SIREs [12] (Fig 1D). The algorithm identified 3638 sequences of interest (S6 Table), based on homology to known IREs. Many were assigned a 'Low' quality score and were not considered further, leaving a total of 1102 genes with IRE-like sequences (S7 Table). Most functionally validated IRE elements in other systems are present in the untranslated regions of transcripts [13]. There were 343 genes with predicted IRE-like sequences in the 5' UTR and 176 genes with IRE-like sequences in the 3' UTR (S8 Table). IREs can be assigned to 18 classes based on sequence conservation to experimentally validated sequences shown to bind aconitase *in vitro*, however the majority of validated IREs are from classes I, II or VIII [12]. As such we focused on the 234 genes with UTR-localised IRE-like sequences from these motif classes (I – 116, II - 92, VIII – 26, S9 Table). We then asked how many of the transcripts containing IRE-like sequences were differentially regulated in our RNA-seq datasets. We found that only 142 (60%) of IRE-like sequence containing genes were regulated by iron availability (padj < 0.05, S9 Table and Figs 1E and S1B). As such, our list of putative IRE-like sequences likely contains false positives.

In order to test whether the IRE-like sequences can impact gene expression we chose to focus on a class VIII IRE-like sequence (5'- gctgcctccgtgtcagggtagacgagagaaa-3') in the 3' UTR of a previously unstudied gene TGME49_261720 which was conserved between *T. gondii* genomes (S2A Fig). TGME49_261720 encodes an essential ZIP-domain containing protein

[44] and is a putative plasma-membrane localised zinc/iron transporter [39]. For these reasons, we renamed the gene ZFT for **Z**n and **F**e **T**ransporter.

## The *zft* 3' UTR is sufficient to render a reporter gene iron responsive

To determine whether the predicted IRE in the *zft* transcript is sufficient to confer iron-responsivity, we cloned *zft* 3' UTR immediately downstream of the fluorescent protein tdTomato under a constitutive promoter (tubulin-tdTomato-*zft*). We also constructed a control line with tdTomato upstream of the *dhfr* 3' UTR (tub-tdTomato-dhfr) which does not contain an IRE-like sequence and we did not expect to be iron responsive. We integrated these cassettes into the *uprt* locus of RH *T. gondii* parasites constitutively expressing mNeonGreen fluorescence protein in the *ku80* locus from the *sag1* promotor with a *sag1* 3' UTR [4] and confirmed by PCR (S2B–S2D Fig). Interestingly, replacement of the *dhfr* UTR with *zft* resulted in significantly lower tdTomato signal in untreated parasites (S2F Fig), demonstrating the influence of the 3' UTR on protein expression, as previously observed [45].

To determine if the *zft* UTR conferred iron responsivity, we pre-treated host cells for 24 hours with the iron chelator deferoxamine (DFO) before infecting with our reporter and control lines. At 24 hours post infection we collected cells and quantified mNeonGreen and tdTomato fluorescence by flow cytometry (Fig 2A and 2B). DFO treatment did not significantly affect mNeonGreen fluorescence in either line and had no effect on tdTomato fluorescence in the tdTomato$_{dhfr}$ line. However, DFO treatment led to a 2-3-fold increase in tdTomato fluorescence in the tdTomato$_{zft}$ line (Fig 2B). We quantified the ratio of tdTomato/mNeonGreen fluorescence and normalised to the untreated lines, this showed a significant ($p < 0.0001$, one way ANOVA) increase in the ratio in DFO treated parasites, compared to the tdTomato$_{dhfr}$ line (Fig 2C). Interestingly, there was no significant difference in the fluorescence ratio in parasites treated with excess iron, demonstrating that this response is only seen in low iron conditions. To confirm these results, we also assessed tdTomato fluorescence by microscopy and confirmed that there was an increase in the ratio of red:green fluorescence in the tdTomato$_{zft}$ line upon DFO treatment, but not in the tdTomato$_{dhfr}$ line (Fig 2D and 2E).

Given the impact of the *zft* 3' UTR on gene expression we chose to test a second 3' UTR with a predicted IRE-like sequence. For this we chose another gene involved in parasite iron metabolism ISU1 (TGME49_237560) the mitochondrial FeS pathway scaffold protein [2]. *isu1* is predicted by SIREs to have a medium quality, class 13, IRE-like sequence in the 3' UTR (5'- catgctgaatgcatgccatgtaccatgaacat -3'). As above we produced an *isu1* reporter line (tub-tdTomato-isu1) and confirmed genomic integration by PCR (S2D Fig) and flow cytometry (S2E Fig) where it showed an intermediate level of fluorescence. We then quantified the ratio of tdTomato fluorescence in untreated and DFO treated tdTomato$_{isu1}$ parasites as above. While we saw a slight increase in tdTomato fluorescence upon DFO treatment (Fig 2F) this change was not statistically significant at this time point (Fig 2G). Other than a difference in the nucleotide sequence between the *zft* and *isu1* IRE-like sequences, we also noted that they occupy different positions in the respective 3' UTRs. The *zft* IRE-like sequence is relatively central in the UTR (position 1016/2144 bp), whereas the *isu1* sequence starts closer to the 3' end of the UTR (position 1149/1278 bp). To examine whether IRE position may be important for function, and therefore offer a hypothesis as to the difference in responsiveness between the *zft* and *isu1* reporters, we plotted the distance from the stop codon of predicted IRE-like sequences in 3' UTRs against whether that transcript was differentially abundant in our RNAseq dataset (Fig 2H). From this we observed that transcripts with IRE-like sequences between 600–1400 bp from the stop codon were more likely to be differentially abundant in iron deprived parasites compared to the control. Therefore,

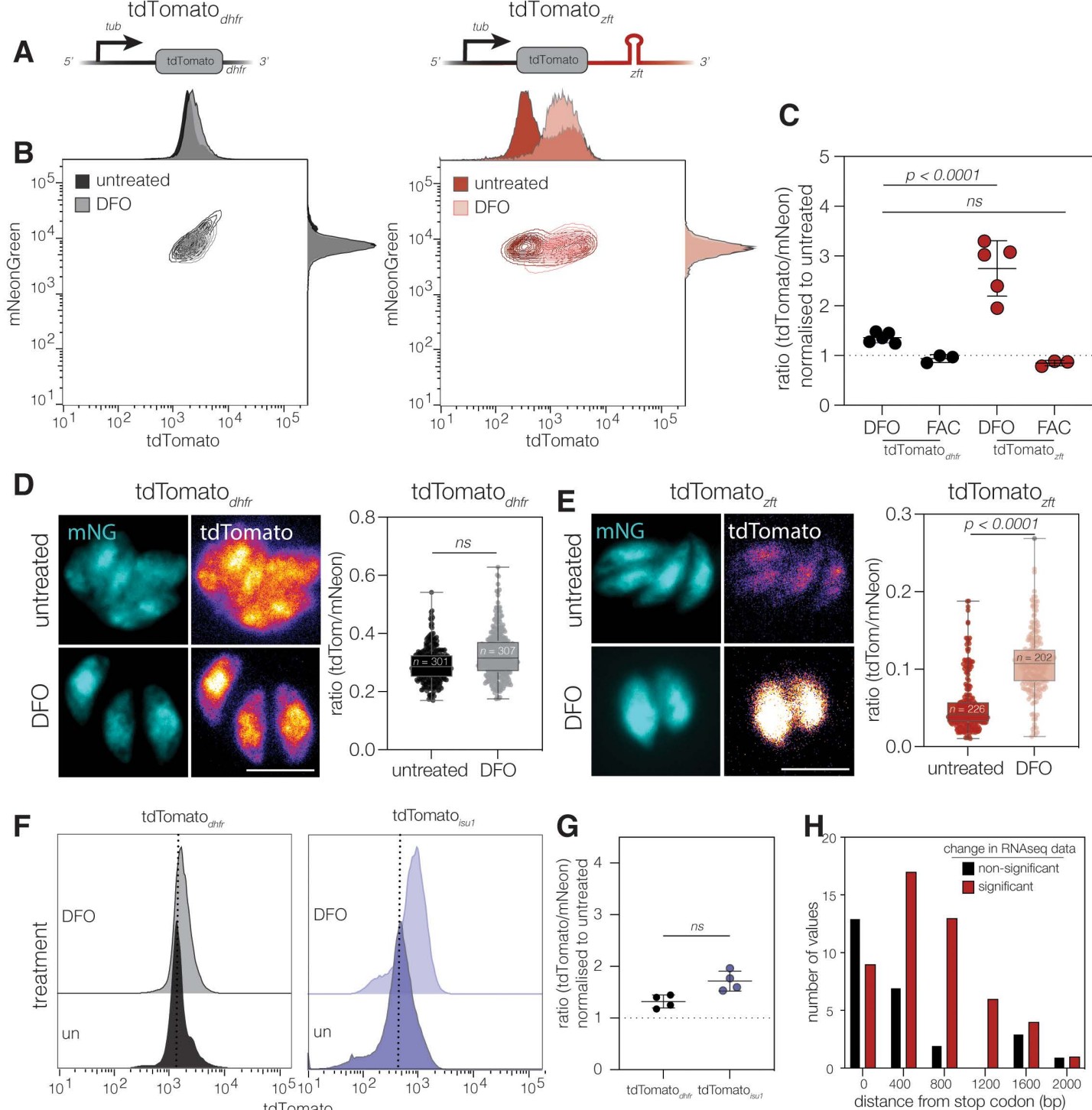

**Fig 2. The *zft* 3' UTR is sufficient to render a reporter gene iron responsive. A**. Schematic of the tdTomato$_{dhfr}$ cassette and tdTomato$_{zft}$ cassette. Both reporters were integrated into the *uprt* locus in a ΔKu80:mNeonGreen parasite line. **B.** Contour plots of mNeonGreen and tdTomato expression in untreated and treated with 100 μM DFO for 24 hours (measured by flow cytometry) for both reporter lines. Representative of at least 5000 parasites from a single experiment. **C.** Plot showing the mean tdTomato:mNeonGreen fluorescence ratio in tdTomato$_{dhfr}$ and tdTomato$_{zft}$ parasites treated with either 100 μM DFO or 2.5 mM ferric ammonium citrate (FAC), normalised to untreated parasites. Plot shows data from 5 independent experiments. *p* values compared to tdTomato$_{dhfr}$ by one way ANOVA with Dunnett's correction **D.** Immunofluorescence showing mNeonGreen and tdTomato expression in untreated and DFO treated (24 hours, 100 μM DFO) tdTomato$_{dhfr}$ parasites with a violin/box plot showing the quantification of tdTomato:mNeonGreen fluorescence from vacuoles from two biological replicates (*n* indicated on graph), *p* values from unpaired t

test. Scale bar 5 μm **E.** Immunofluorescence showing mNeonGreen and tdTomato expression in untreated and DFO treated (24 hours, 100 μM DFO). Scale bar 5 μm. tdTomato$_{zft}$ parasites with a violin/box plot showing the quantification of tdTomato:mNeonGreen ratio as above. **F.** Representative overlapping histograms showing tdTomato fluorescence in the tdTomato$_{isu1}$ reporter line both untreated and after 24 hours of treatment in 100 μM DFO. **G.** Plot showing the mean tdTomato:mNeon-Green fluorescence ratio in untreated tdTomato$_{isu1}$ and those treated with 100 μM DFO, normalised to untreated parasites. Data shown from 4 independent experiments. *p* value from unpaired t test. **H.** Histogram showing the frequency of differentially regulated transcripts with IRE-like sequences in their 3' UTRs in RNAseq data. Transcripts were binned (bin size = 400 bp) based on distance of the IRE-like sequence from the stop codon.

IRE-like positioning may be important to function, and this may account for the lack of response seen in the tdTomato$_{isu1}$ line.

## *zft* UTR responsiveness is DFO dose dependent and not responsive to zinc or copper chelation

Given the responsiveness of the tdTomato$_{zft}$ line, we then further validated the reporter line to determine whether this response is dosage dependant. We first repeated our initial experiments by pre-treating host cells with concentrations between 0–500 μM of DFO for 24 hours before infection with the tdTomato$_{dhfr}$ and tdTomato$_{zft}$ reporter parasites for 24 h, then quantifying tdTomato fluorescence by flow cytometry (Fig 3A). As previously observed, tdTomato fluorescence in the tdTomato$_{dhfr}$ parasite line was not significantly altered by any tested concentration of DFO, however we observed a dose-dependent increase in signal in tdTomato$_{zft}$ parasites (Fig 3B). Following these experiments we produced a second *zft* reporter line, this time cloning the 3'UTR downstream of nanoLuciferase (nanoLuc$_{zft}$) (S2G Fig) – a reporter gene with improved dynamic range and lower reported toxicity [46]. We confirmed nanoLuciferase expression (S2H Fig) and then repeated the dose-response experiment in the new reporter. We find that the nanoLuc$_{zft}$ reporter was also sensitive to iron depletion in a similar range to the tdTomato$_{zft}$ reporter line (Fig 3C).

We then asked whether the tdTomato$_{zft}$ reporter responses are consistent when treated with different iron chelators. To this end we treated both tdTomato$_{dhfr}$ and tdTomato$_{zft}$ parasites with two further membrane-permeant iron chelators, the $Fe^{2+}$ chelator, 2,2'-bipyridine (BIP) and the $Fe^{3+}$ chelator pyridoxal isonicotinoyl hydrazone (PIH). Interestingly, while the response to BIP was indistinguishable from DFO (Fig 3D and 3E), there was no change in tdTomato fluorescence upon treatment with PIH. To investigate this further, we quantified the effectiveness of the three chelators in inhibiting parasite growth (Fig 3F) and found that DFO showed the greatest inhibition of parasite replication ($EC_{50}$ 8.68 μM (95% C.I. 6.63–11.31 μM)), with BIP also showing high potency ($EC_{50}$ 17.05 μM, 95% C.I. 11.9 – 24.13 μM) while PIH was significantly (*p* = 0.046, Sum of squares F test) less able to prevent parasite replication ($EC_{50}$ 121.01 μM, 95% C.I. 97.6 – 148.3 μM), suggesting that it is less able to remove parasite-available iron than DFO or BIP, under these conditions.

Given that several metal iron transporters are able to transport multiple metal species, e.g., the *Plasmodium* ZIPCO likely has roles in the transport of both iron and zinc [47], we sought to confirm that the response seen was specifically due to removal of iron, we also tested two other metal chelators, *N,N,N',N'*-tetrakis-(2-pyridylmethyl)ethylenediamine (TPEN), a broadly active metal ion chelator with high affinity for $Zn^{2+}$ [48] and Tetrathiomolybdate (TTM), a selective copper chelator [49]. Treatment with either of these chelators did not lead to an increase in the tdTomato:mNeonGreen ratio (Fig 3G and 3H), suggesting that the response seen is specific to iron chelation and is not a generalised response upon removal of other metals.

## Kinetics of iron response

To determine the kinetics of the response to iron in our reporter line, cells pretreated with DFO were infected for the indicated times before parasites were mechanically removed.

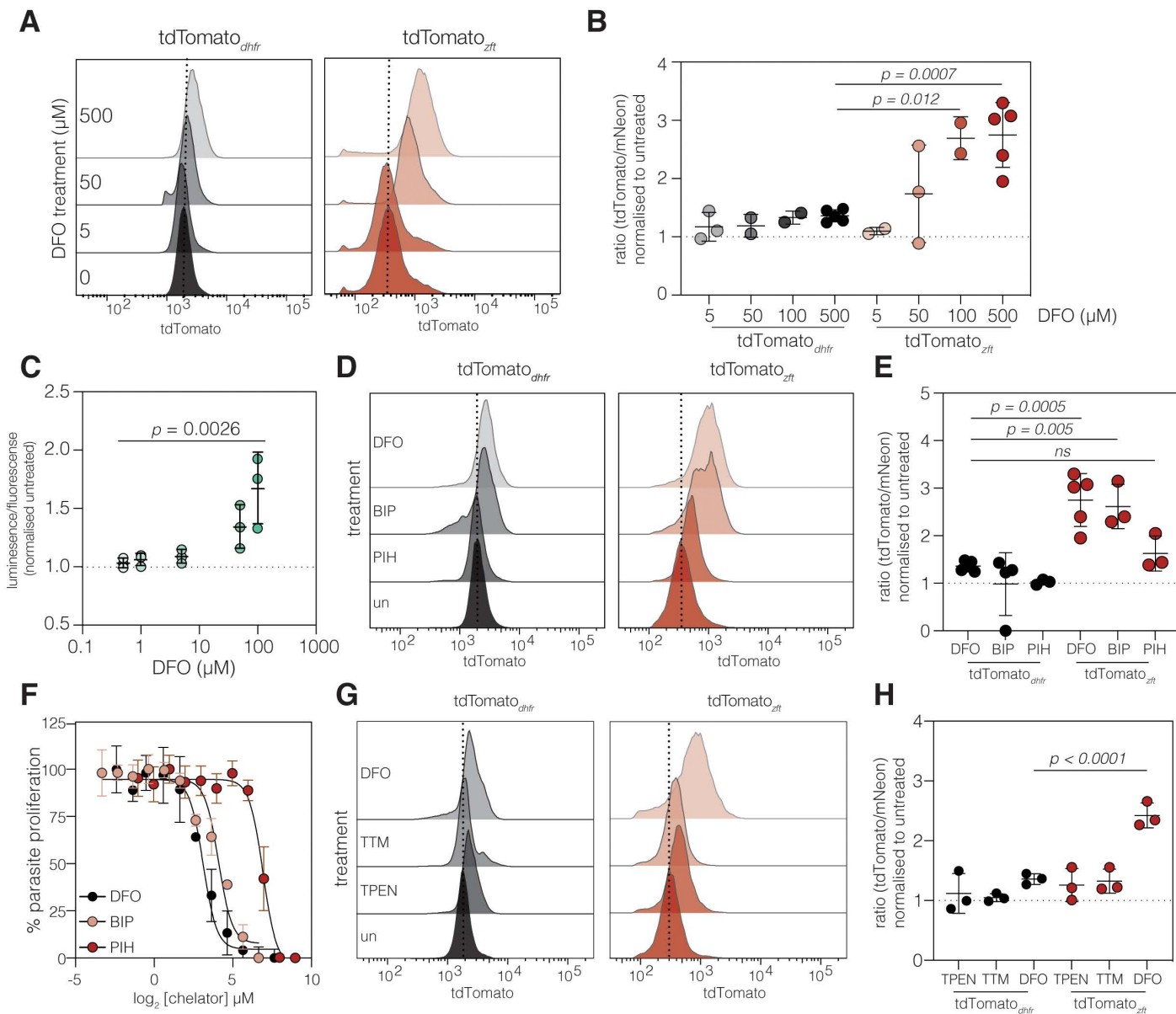

**Fig 3. ZFT responsiveness is dose dependent and specific to iron chelation. A.** Representative overlapping histograms showing tdTomato fluorescence in the tdTomato$_{dhfr}$ and tdTomato$_{zft}$ reporter lines treated with indicated DFO concentration for 24 hours. **B.** Quantification of change of tdTomato:mNeonGreen ratio. *p* values from one way ANOVA with Dunnett's correction, compared to tdTomato$_{dhfr}$ + 500 μM DFO **C.** Quantification in luminescence signal (normalised to parasite flourescense and untreated cells) after treatment with indicated DFO concentration for 24 h. *p* values from one way ANOVA with Dunnett's correction, compared to parasites treated with 0.5 μM DFO **D.** Representative overlapping histograms showing tdTomato fluorescence, measured by flow cytometry, in the tdTomato$_{dhfr}$ and tdTomato$_{zft}$ reporter lines treated with 100 μM DFO, 100 μM BIP or 100 μM PIH for 24 hours. **E.** Mean tdTomato:mNeonGreen fluorescence ratio for three independent experiments. *p* values from one way ANOVA with Dunnett's correction, compared to tdTomato$_{dhfr}$ + DFO. **F.** Fluorescence growth assay for tdTomato parasites cultured in increasing concentrations of iron chelators. Points show mean of three independent experiments ± SEM. **G.** Representative overlapping histograms showing tdTomato fluorescence, measured by flow cytometry, in the tdTomato$_{dhfr}$ and tdTomato$_{zft}$ reporter lines treated with 100 μM DFO, 5 μM TPEN (Zn chelator) or 25 μM TTM (Cu chelator) for 24 hours. **H.** Mean tdTomato:mNeonGreen fluorescence ratio for three independent experiments. *p* values from one way ANOVA with Dunnett's correction, compared to tdTomato$_{dhfr}$ + DFO.

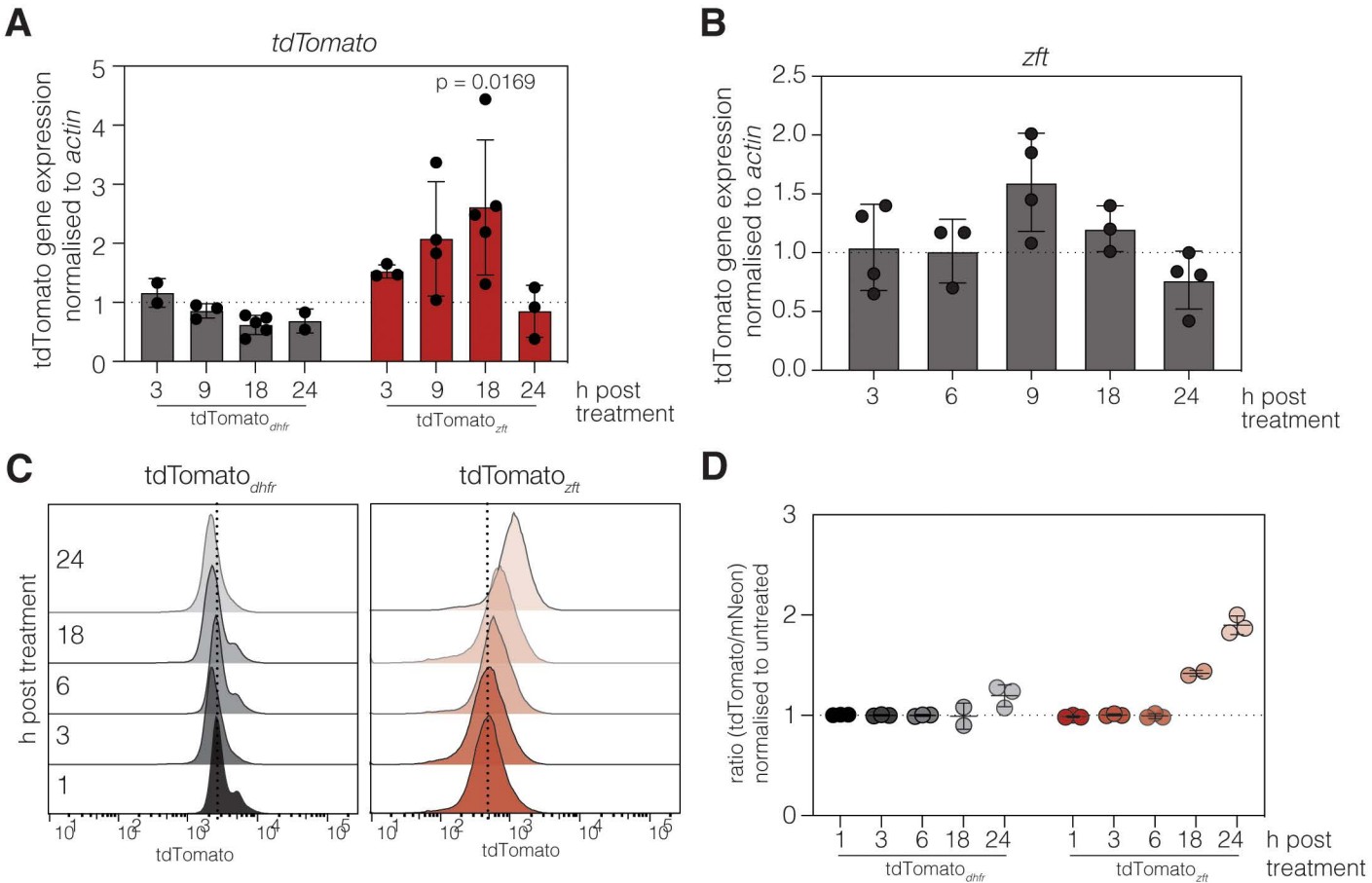

**Fig 4. Kinetics of tdTomato*zft* iron-responsiveness.** qRT-PCR on the relative abundance of *tdTomato* (**A**) and *zft* (**B**) transcripts (normalised to actin) in the tdTomato$_{dhfr}$ and tdTomato$_{zft}$ reporter lines after treatment with 100 μM DFO, compared to untreated parasites for indicated time. Points represent a single experiment, bars at mean ± SD. **C.** Representative overlapping histograms showing tdTomato fluorescence, measured by flow cytometry, in the tdTomato$_{dhfr}$ and tdTomato$_{zft}$ reporter lines treated with 100 μM DFO over a 24-hour time course. **D.** Plot showing the mean tdTomato:mNeonGreen fluorescence ratio for the experiments described in (**C**).

To determine the timing of the response at the transcript level, we performed qRT-PCR on reverse transcribed RNA taken from the parasites (Fig 4A). We saw no significant change in the tdTomato transcript abundance (relative to actin) in the tdTomato$_{dhfr}$ cells. However, in the tdTomato$_{zft}$ line there was an increase in transcripts detectable from 9 hours ($p = 0.0849$, mixed effects test with Tukey's multiple comparisons test), which then was significantly increased at 18 hours of treatment ($p = 0.0169$). However, by 24 h (the peak of tdTomato protein abundance) mRNA levels had fallen back to untreated levels. This pattern in regulation was similar to that of the endogenous *zft* transcripts (Fig 4B), though the magnitude of the increasing transcript abundance was more modest – possibly reflecting the difference in promoter strength. We also examined protein expression over the same time course using flow cytometry (Fig 4C). We did not observe any significant change in tdTomato fluorescence in the tdTomato$_{dhfr}$ cells at any timepoint. However, in the tdTomato$_{zft}$ line, we saw a gradual increase in tdTomato signal, starting at 18 h post infection with a further increase at 24 h, later than the peak of increased transcript level. These results show that mRNA levels peak prior to protein, as would be expected. It also suggests that this response to iron deprivation relies on new protein synthesis, rather than an inhibition of protein degradation.

## Removal of the IRE-like element from the *zft* 3' UTR reduces iron responsiveness

To determine if the predicted IRE sequence was important for conferring iron-responsivity in the *zft* 3' UTR, we deleted an 18 nucleotide sequence from the predicted IRE (bases 2–25 of the predicted sequence), to create a tdTomato$_{zft\Delta IRE}$ line (S2D, S2H and 5A Figs). Deletion of the IRE has previously been shown to abolish IRE function in other transcripts [50,51]. Using this line, we assayed transcript abundance in the tdTomato$_{zft\Delta IRE}$ reporter line at 18 hours of treatment, the peak of response from the above time course (Fig 4A). The tdTomato$_{zft}$ line had significantly more tdTomato transcripts ($p$ = 0.0019, one way ANOVA with Tukey's correction) than the tdTomato$_{dhfr}$ line (Fig 5B). However, tdTomato transcript abundance for the tdTomato$_{zft\Delta IRE}$ was not significantly different from the tdTomato$_{dhfr}$ line ($p$ = 0.44, one-way ANOVA with Tukey's correction). We also confirmed this at the protein level at 24 h post infection. As shown above, the presence of the *zft* UTR induced a strong increase in tdTomato expression (Fig 5C), compared to the control tdTomato$_{dhfr}$ line ($p$ = 0.0023, one-way ANOVA with Dunnett's multiple comparisons test) (Fig 5D). Deletion of the IRE-like element (while not affecting total tdTomato fluorescence in untreated cells (S2H Fig)) dampened this responsiveness and was not significantly different from the tdTomato$_{dhfr}$ line. This suggests that the presence of the IRE-like element modulates the iron-responsiveness of the tdTomato$_{zft}$ reporter, although potentially other structures in the IRE also play a role in mediating the response.

Given the association of iron stress and genes associated with bradyzoite conversion we also measured reporter response to alkaline stress conditions known to induce parasite differentiation *in vitro* [35]. At 24 h post induction we saw the start of bradyzoite differentiation with accumulation of the bradyzoite marker BAG1 (S2I Fig). As such, in accordance with the expression of bradyzoite differentiation markers, we analysed our reporter lines at 24 h post induction. We find that 24 h of alkaline stress is sufficient to induce protein accumulation in the tdTomato$_{zft}$ line compared to the control tdTomato$_{dhfr}$ line (Fig 5F) although the response was more muted compared to iron chelation. The response in the absence of the IRE was similar. These results fit with the RNAseq data demonstrating that iron depletion leads to induction of a bradyzoite program.

## Iron deprivation alters transcript stability

Iron-mediated changes in transcript stability have been documented from mammals [52] to yeast [53], and so we sought to determine the effect of iron deprivation on mRNA stability in *Toxoplasma*. To this end, we measured transcript abundance after blocking new transcription with actinomycin D, as previously described [54]. Transcript levels of tdTomato$_{dhfr}$ fell over the time course, in both untreated and low iron conditions, reaching approximately 17% of the starting level after 5 hours of treatment with no significant difference upon treatment ($p$ = 0.19, extra sum of squares F-test) (Fig 6A). tdTomato$_{zft}$ showed a similar pattern in untreated cells, with a more rapid transcript depletion followed by a plateau between 1 and 5 h. However, under low iron conditions we saw a significant ($p$ < 0.0001, extra sum of squares F-test) stabilisation of the tdTomato$_{zft}$ transcript with transcript abundance remaining around 100% even after 5 hours of actinomycin D treatment (Fig 6B). We also measured endogenous *zft* stability (Fig 6C) and saw that DFO-treatment significantly ($p$ = 0.01, extra sum of squares F-test) stabilised the native transcript. We believe that this stabilisation likely leads to the increased mRNA abundance we observed, and later to an increase in protein levels (Fig 4).

Following these observations, we assayed the iron-dependent transcript stability for other selected transcripts with differential expression by RNAseq (Fig 1E) and with putative

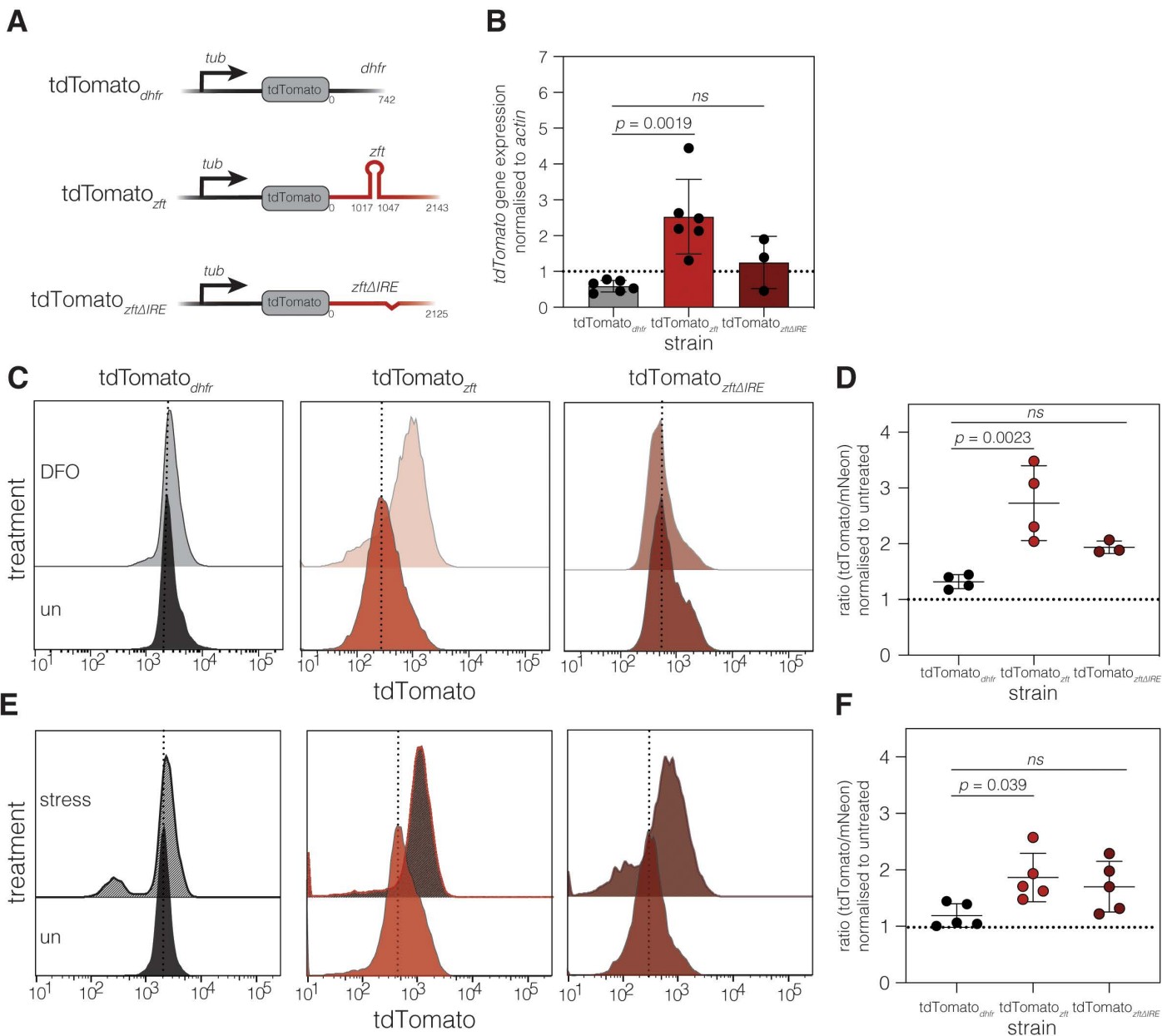

**Fig 5. Removal of the IRE-like element from the *zft* 3' UTR reduces iron responsiveness. A.** Schematic of the iron reporter cassettes integrated into the *uprt* locus. **B.** qRT-PCR on the relative abundance of *tdTomato* transcripts (normalised to actin) in the tdTomato$_{dhfr}$, tdTomato$_{zft}$ and tdTomato$_{zft\Delta IRE}$ reporter lines after 18 hours treatment with 100 μM DFO, compared to untreated parasites. Bars at mean ± SD, *p* values from one way ANOVA with Tukey's correction. Bars at mean ± SD, *p* values from one way ANOVA with Tukey's correction. **C.** Representative overlapping histograms showing tdTomato fluorescence in the tdTomato$_{dhfr}$, tdTomato$_{zft}$ and tdTomato$_{zft\Delta IRE}$ reporter lines, treated with 100 μM DFO for 24 hours. **D.** Mean tdTomato:mNeonGreen fluorescence ratio, *p* values from one way ANOVA with Dunnett's correction, compared to tdTomato$_{dhfr}$ + DFO. **E.** Representative overlapping histograms showing tdTomato fluorescence in the tdTomato$_{dhfr}$, tdTomato$_{zft}$ and tdTomato$_{zft\Delta IRE}$ reporter lines, treated with alkaline stress for 24 hours. **D.** Mean tdTomato:mNeonGreen fluorescence ratio, *p* values from one way ANOVA with Dunnett's correction, compared to tdTomato$_{dhfr}$ + stress.

IREs: *isu1*, *Tg253510* (a predicted transporter) and *Tg500296* (a KRUF family protein). We also included *sag1* as a negative control, based on the lack of changes in the mNeonGreen protein levels (Fig 2A). While we saw no change in stability for *sag1* (as expected) or *Tg500296*, we observed significantly (extra sum of squares F test) increased stability in both *Tg253510*

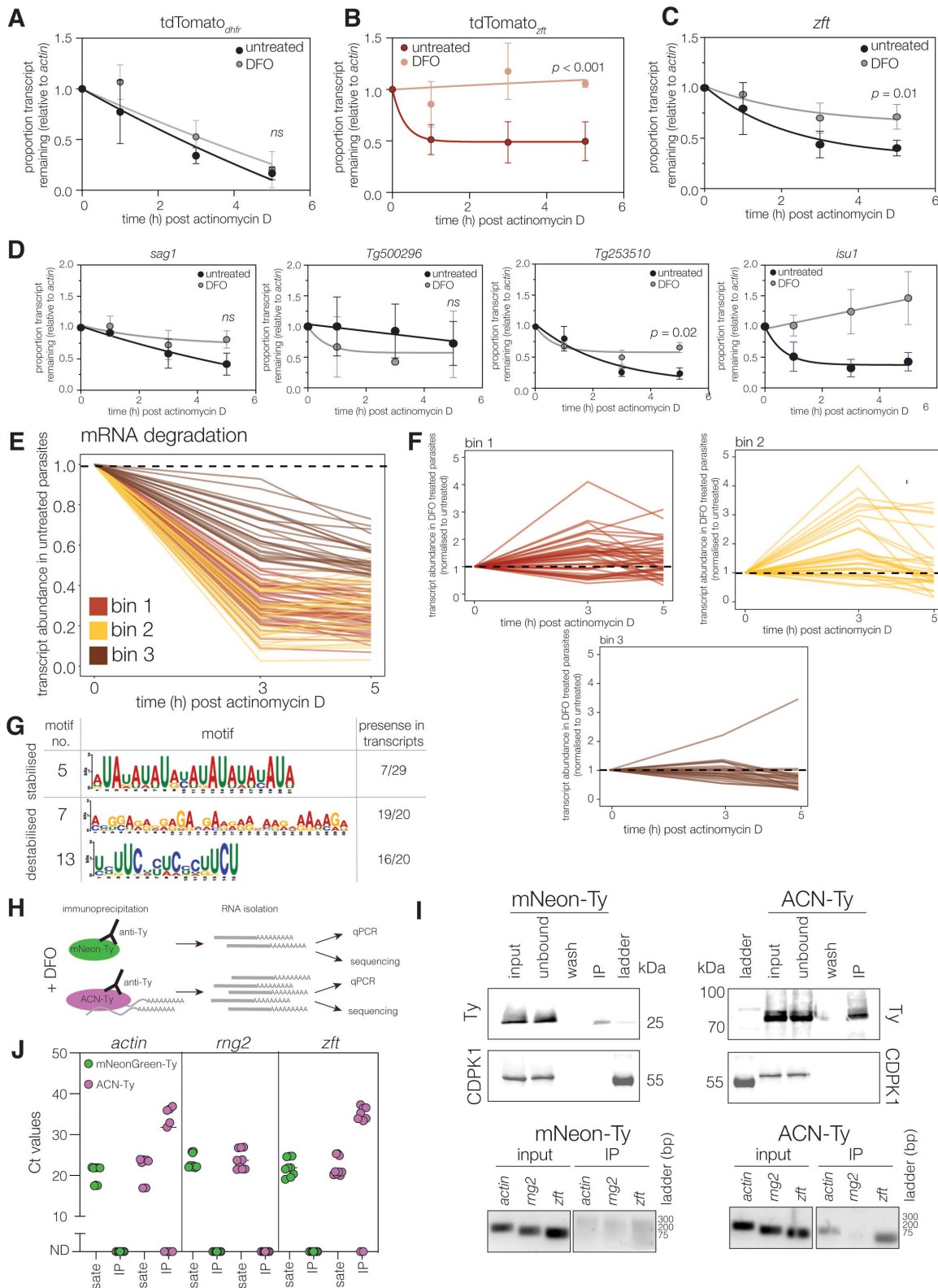

**Fig 6. *zft* 3'UTR demonstrates iron-dependent mRNA stability.** Untreated or DFO-treated tdTomato_{dhfr} (**A**) or tdTomato_{zft} (**B**) or parental parasites (**C**) treated intracellularly with actinomycin D and qPCR performed for indicated gene. Results normalised to *actin*, results mean of three independent replicates, ± SD, *p* value from extra sum of squares F test. **D.** mRNA stability performed as above,

targeting *sag1*, *Tg500296*, *Tg253510* or *isu1*, normalised to *actin*. Results mean of three independent replicates, ± SD, *p* values from extra sum of squares F test, due to the lack of degradation for *isu1*, we were unable to calculate a *p* value. **E.** Average abundance for transcripts which reached <75% of t = 0 across a 5-hour time course of actinomycin D treatment from untreated parasites. Transcripts were binned based on decay kinetics. Bin 1- 50% abundance by 3 hours, then further decay; Bin 2 - 50% abundance by 3 hours, without further decay and Bin 3 – transcripts which did not decay >50% during the time course **F.** Binned average transcripts remaining from DFO-treated parasites, normalised to parasites cultured in standard conditions. Binning information above. **G.** Selected motifs significantly enriched in DFO-responsive transcripts (from XSTREME), with proportion of transcripts containing the motif indicated. **H.** Schematic showing protocol for aconitase IPs. **I.** Blots showing the immunoprecipitation of mNeonGreen-Ty and ACN-Ty from *T. gondii* parasites. CDPK1 included as cytosolic control. Wash – the output of the 6th and final washing step. Representative of three independent replicates. DNA-agarose gel showing qPCR products amplified from reverse transcribed RNA from either starting lysates or RNA co-precipitated with mNeonGreen-Ty and aconitase-Ty proteins. PCR amplicons for *actin* (expected size: 188 bp), *zft* (expected size: 116 bp) and *rng2* (expected size: 146 bp). **J.** qRT-PCR results from IPs performed in (**H**). Points represent average Ct values from individual biological replicates, performed in triplicate, lines at median. ND – not detected.

and *isu1* transcripts (Fig 6D). This was surprising as the tdTomato$_{isu1}$ reporter line did not significantly respond to iron deprivation at the same timepoint (Fig 2F and 2G), which may suggest that additional genomic context is required for regulation of some transcripts, or that the peak of protein changes occurs at a different time point which we did not assay.

Given that iron availability impacted the stability of the *zft*, *isu1* and *Tg253510*, but not *Tg500296* or *sag1*, transcripts, we sought to examine mRNA stability across the transcriptome. We repeated the stability assay using RNA-seq to compare the decay of transcripts in DFO-treated to those of untreated cells. PCA analysis demonstrated good agreement between replicates (S3A Fig). For our analyses we filtered out transcripts which did not decay within the experimental timeframe using a cut-off of <75% of transcripts remaining after 5 hours of actinomycin D treatment, relative to the untreated control (one standard deviation from the mean transcript abundance at 5 hours with an adjusted p-value of <0.05). This resulted in 426 transcripts (S12 Table). To aid inspection of the data, we divided these transcripts into three bins based on their decay kinetics under actinomycin treatment alone (Fig 6E). Bin 1 contains genes which reach 50% abundance by 3 hours treatment and decay further at 5h, bin 2 includes genes which reach 50% transcript abundance by 3 hours treatment but did not decay further, and bin 3 contains genes which did not reach 50% abundance by 3 hours of treatment. We then normalized the transcript abundance in DFO treated parasites to that of parasites treated with actinomycin D alone (Fig 6F). From this, we identified 29 transcripts that were twice as stable in DFO treated parasites (scores of > 2) including TGME49_221270, a ferrous iron-dependent oxidoreductase. We also found 20 transcripts which were half as stable during DFO treatment (scores of < 0.5). We did not observe enrichment of genes with predicted IREs from this dataset (6/49, fishers exact test *p* = 0.0648) suggesting that they contain non-canonical iron-regulatory elements or that other properties of these transcripts are important for determining their stability in low iron. As such, we looked for conserved RNA motifs from the differentially stabilized enriched transcripts using the web-based motif discovery and analysis tool XSTREME [55,56]. From the transcripts which showed increased stability we found 6 motifs (S12 Table, Motifs 1-6). Four of the motifs were novel and found in approximately half of the stabilized transcripts. Additionally, an AU-rich motif, similar to known RNA motifs RBMS3 (RNCMPT00173, RNCMPT00057) and SUP-26 (RNCMPT00182) (Motif 5, 5'- AUAUAUAUAYAUAUAUAUAUA -3') was found in 7/29 transcripts (Fig 6G). It is known that AU-rich elements (AREs) play a role in the iron regulation of yeast through the action of RBP Cth2 [57,58]. There is no clear Cth2 orthologue in *T. gondii*, however there are 56 proteins which share the zinc finger CCCH domain required for Cth2 to function. From the transcripts which were less stable following DFO treatment, we found that 19/20 transcripts contained motifs resembling those of poly(A) binding proteins N1 and C (Motif 7). Additionally we found 5 novel motifs. The most common of which was found

in 16/20 of the destabilized transcripts (Motif 13, 5'-UYUUCWYUCYCUUCU-3') (Fig 6G). None of the discovered motifs were found in all transcripts of interest in either group, as such their significance in the context of iron remains unclear. Future work will uncover if any are responsible for iron mediated changes to stability.

## Aconitase mRNA binding in low iron conditions

Aconitase binding to IRE-sequences in transcripts has been shown to lead to stabilisation of the mRNA, resulting in increased protein production [23,59,60]. The single aconitase of *T. gondii* (ACN, TGME49_226730) shows high sequence similarity (S4A Fig) and predicted structure (S4B Fig) with the cytosolic mammalian aconitase, including key residues known to be important for binding the catalytic iron-sulphur cluster and RNA-binding [61]. To investigate the role of aconitase in the response to iron, we made use of a parasite line with aconitase tagged with a Ty epitope (ACN-Ty) (S3B Fig). ACN-Ty localised to the mitochondrion, apicoplast and cytosol as previously described [62], and localisation was not significantly affected by iron depletion (S3C Fig). It has previously been reported that aconitase requires FeS-binding to enable enzymatic activity [63–65]. To determine if iron depletion affected *T. gondii* aconitase enzymatic function, activity was quantified by NADPH accumulation [66]. Iron depletion by DFO treatment led to a small, but reproducible and significant reduction, in enzymatic activity ($p < 0.0001$, paired t-test, S3D Fig), as indicated by a decrease in NADPH in treated parasites. A control of filtered, uninfected host cells showed no measurable host cell aconitase activity.

Aconitase presence in the cytoplasm would potentially allow the protein to interact with mRNAs. Given this, and the conservation of aconitase RNA-binding in several clades, we investigated whether *T. gondii* aconitase can bind mRNA (Fig 6G). To this end, parasite ACN-Ty was immunoprecipitated from parasites cultured in a low iron environment for 24 hours. As a control, mNeonGreen-Ty [4] was also immunoprecipitated (Fig 6H). RNA was then extracted from elutants (and parasite lysates as a loading control) and reverse transcribed to allow identification of associated transcripts by qPCR. Here, we included *actin* as a highly transcribed mRNA and *rng2*, a structural protein of the apical complex [67], as a control transcript which has a comparable expression to *zft* within our transcriptomics dataset, but did not show altered expression (S3E Fig) and did not contain a predicted IRE. Protein-associated cDNAs were quantified by qRT-PCR (S10 Table) and visualised on a DNA agarose gel (Figs 6C, S3F and S3G). From ACN-Ty-associated cDNAs we detected transcripts by qRT-PCR from both *actin* and *zft*, but not *rng2* (Fig 6H and S8 Table) and we were unable to detect any transcripts associated with mNeonGreen-Ty. To investigate specificity of this potential interaction, we sequenced all RNAs which co-immunoprecipitated with either ACN-Ty, or mNG-Ty. These experiments did not show enrichment of the *zft* transcript in the ACN-Ty pulldown, compared to starting lysate. We did however find 43 transcripts significantly (padj < 0.05) enriched in the ACN-Ty pulldown (compared to starting lysates) and not in the mNG-Ty pulldown (S11 Table). We do not find any enriched GO terms or metabolic pathways in this geneset, however we did find that these genes are significantly enriched for transcripts containing a putative IRE (13/43, Fishers exact test, $p < 0.0001$). It is also possible that the remaining transcripts contained non-canonical IREs which were not predicted. As above, we looked for other conserved motifs from the 43 enriched transcripts using XSTREME [55,56]. This search identified five enriched motifs – three similar to known motifs from other eukaryotes and two novel motifs (S11 Table). The most common enriched motif (4) was found in 36/43 transcripts and is similar to the motifs of poly-A associated RBPs including PABPN1 [68], REF2 [69]. Two further motifs, motif 3 (33/43 transcripts, 5'- GCGUCGGACGCGCCK-3') and motif 5 (21/43 transcripts,

5'- AUMYAUAUAUAUAUAUAUAUAUAUAUAUAUAU-3'), are predicted to form stem-loop structures by RNAfold [70], though with half the loop size of the *zft*-predicted IRE (10 bases). However, as none of the motifs were found in all transcripts of interest, it is currently unclear whether any could provide an alternative ACN binding site.

Together these results do not suggest a specific interaction between ACN and the *zft* transcript in *T. gondii*, although we do not rule out a potential role for ACN in binding and modulating other IRE-containing transcripts. These results mean the potential identity of the *zft*-transcript binding protein remains unknown.

### The ZFT 3'UTR is important for parasite fitness in low iron

Following these experiments, we examined the impact of the 3'UTR on native protein level expression of ZFT itself. To this end we c-terminally tagged ZFT at the endogenous locus with 3xHA in two lines, one with the endogenous 3'UTR (ZFT-HA$_{zft}$) and another with the *sag1* UTR (ZFT-HA$_{sag1}$, S5A and S5B Fig) which we have established is not iron regulated and does not confer differential mRNA stability. The lines were then treated with DFO, and ZFT protein abundance quantified by western blot. In the ZFT-HA$_{sag1}$ line we observed that ZFT protein levels were significantly reduced at both 18 h and 24 h of DFO treatment (Fig 7A and 7B). However, in the presence of the endogenous UTR line we saw that ZFT levels were largely (80% of the untreated) maintained at 18 h and then reduced by 24 h iron deprivation. This suggests that the endogenous 3'UTR stabilizes ZFT proteins levels initially upon reduction of available iron.

Finally, we sought to determine whether the observed dysregulation of ZFT in the absence of its native 3'UTR impacts parasite growth. We found no change in the size of plaques in untreated cells between parental parasites, those with HA-tagged with the native UTR or HA tagged with the *sag1* UTR (S5 Fig) under native conditions. To determine the effects when iron is limiting, we infected cells in a 96 well plate with increasing concentrations of DFO. After four days we stained the monolayer with crystal violet and quantified the remaining host cell monolayer by absorbance. We observed that ZFT-HA$_{sag1}$ parasites were significantly (extra sum of square F-test $p < 0.0001$) more sensitive to DFO treatment (EC$_{50}$ 6.5 μM) than either line containing the native UTR (parental EC$_{50}$ 18.6 μM, ZFT-HA$_{zft}$ EC$_{50}$ 16.5 μM) (Fig 7C). To further confirm this effect on survival, we treated cells with a low concentration of DFO (1 μM) which we found inhibited plaque area by around 30% (Fig 7D). Under these conditions, we saw a mild, but significant ($p = 0.038$, one way ANOVA) decrease in plaque area in parasites with the non-native *zft* UTR (Fig 7E). These data demonstrate the importance of ZFT regulation through its 3'UTR for parasite growth in a low iron environment.

### Discussion

Here we show for the first time the existence of iron-mediated post-transcriptional responses in *T. gondii.* The ability of cells to respond to iron availability is conserved across the tree of life, however the mechanistic details of regulation are often species-specific. In *T. gondii*, iron deprivation leads to a substantial transcriptomic upregulation, with over 13% of the genome significantly upregulated (log$_2$ fold change > 2), and few genes down-regulated. Along with changes in genes predicted to encode iron-bound proteins and their synthesis pathways, we see evidence that iron deprivation leads to an upregulation of genes associated with the bradyzoite stage (e.g., BAG1, LDH2 and ENO1) [35]. Interestingly, repeating the experiment in the absence of the master regulator of stage conversion, BFD1, shows sustained upregulation of many BFD1-dependent factors, although to a lesser degree. This suggests that although BFD1 is sufficient for stage conversion, and parasites cannot

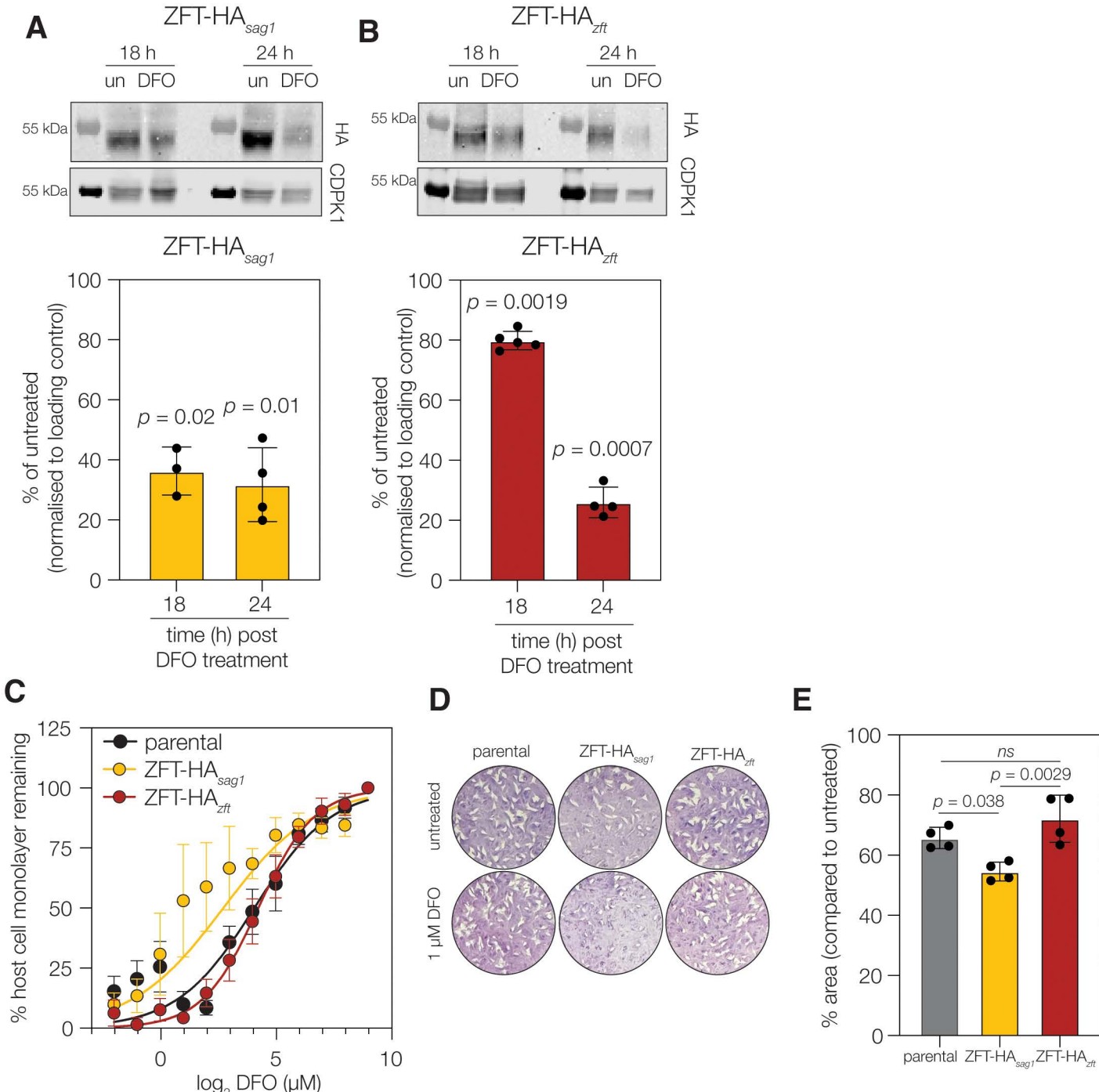

**Fig 7. Presence of *zft* 3' UTR affects native protein. A.** Blots showing ZFT-HA*sag1* protein levels after 18 and 24 h treatment with DFO. Protein levels are significantly (t test, compared to untreated) reduced at both time points. Bars at mean ± SD, points represent values from independent biological experiments. **B.** ZFT-HA*zft* parasites tagged with the native UTR showed significant reduction at 24 h of treatment, but protein levels were maintained at 18 h. **C.** Host cell monolayer absorbance after infection in the presence of increasing DFO. Points show mean of four independent experiments ± SEM. **D.** Representative plaque assays after 7 days of growth either untreated or in the presence of 1 µM DFO. **E.** Plaque area quantification showing % change in plaque area compared to untreated. Bars at mean ± SD and points represent mean plaque area from four independent experiments. *p* values from one way ANOVA with Tukeys correction.

fully differentiate in its absence, other signalling pathways can also lead to initiation of bradyzoite gene transcription. Our results, combined with other recent data [71], confirms that iron deprivation joins an extensive list of cellular stresses able to trigger stage conversion in *Toxoplasma*. However, the signalling pathway of iron starvation leading to differentiation remains unknown.

The lack of apparent specificity in the transcriptional response suggested to us that there were further, post-transcriptional layers of regulation to control *T. gondii's* response to iron deprivation. Using a sequence-based search algorithm [12], we were able to identify genes containing iron response element-like sequences. One limitation of this approach is that these predictions are made based on data from human and mouse cells; *T. gondii* are highly divergent from model opisthokonts [72] and so while these predictions are likely to be an overestimate, we cannot exclude that functional but divergent IREs are also present, as previously hypothesised in *Plasmodium* [32]. Additionally, the short sequence-based search algorithm utilised cannot consider the full context of transcripts (e.g., folding or secondary structures) and so some predictions are likely to be false positive. Despite this, we were able to identify TGME49_261720 as containing a putative IRE sequence and were able to validate the UTR conferred iron-responsivity. Interestingly, we also saw a small but significant increase in abundance of our reporter upon alkaline stress. The mechanism for this is not yet understood, however the endogenous gene is not significantly upregulated in bradyzoites [35], suggesting the existence of further layers of regulation for this transcript.

The response to iron depletion seen here was dynamic; mRNA accumulation was detected between 9–18 h post treatment, while protein levels increased between 18–24 h post treatment. We did not extend treatment beyond 24 h, as between 24–48 h we saw a sharp drop in parasite viability. We found that removal of 18 nucleotides from the *zft* 3' UTR mostly abrogated the iron-responsiveness, suggesting that this motif plays a key role. Whilst it is possible that removal of these nucleotides altered the mRNA folding, we did not see significant changes in protein abundance in our untreated reporter lines, mitigating this concern. However, mRNA accumulation was transient, with levels returning to below the untreated by 24 h, explaining the finding that *zft* is significantly downregulated in our 24 h-treated RNAseq (S3E Fig). We do not yet know the mechanisms underlying this repression, although we see that endogenous ZFT-HA protein levels are also repressed at 24 h post treatment.

However, we did find that upon iron deprivation, specific transcripts are stabilised. This includes *zft* and other predicted-IRE containing transcripts (*isu1* and *Tg253510*). We also show, using RNA-sequencing of parasite transcriptomes after transcriptional inhibition, many other transcripts without predicted IREs have altered stability in iron deprivation. This experiment was hampered by many transcripts lacking reads across all samples (as was the case for *isu1*, *Tg253510* and *zft*) which as such resulted in their exclusion from downstream analysis. However, together these data suggest there are other iron-responsive sequences in these transcripts which were not found using existing algorithms. We identified several sequences enriched in differentially stable transcripts using sequence-based search algorithms, future work will explore if any are responsible for iron-mediated RNA stability.

We then sought to determine if the changes in mRNA stability are due to binding to aconitase. In mammalian systems, aconitase binding to IRE-sequences has been shown to stabilise the TfR transcript, leading to increased protein levels [52,73]. *T. gondii* encodes a single aconitase gene, required for the TCA cycle [74,75] which conserves all predicted RNA-binding residues. By immunoprecipitating aconitase from *T. gondii* in low iron conditions, we were reproducibly able to amplify *zft* mRNA and occasionally the highly abundant *actin*. However, by RIP-seq we found that aconitase pulls down numerous transcripts, but not *zft*, making it an

unlikely candidate for the changes in the *zft*-reporter expression. However, the fact that we see an enrichment in genes with predicted IREs in the bound transcripts, does indicate a possible role for aconitase in regulation in *T. gondii*. Previous work from *Plasmodium* has shown PfACN is able to bind to both mammalian IREs and endogenous IREs [32], supporting our RIP-seq results, although the function of this is currently unknown. For *zft*, although our data does not identify a specific role for aconitase in its regulation, we believe a currently unidentified mRNA-binding protein may be responsible.

One important unanswered question in this work is the role of ZFT. ZFT encodes a zinc and iron permase (ZIP) domain containing protein which is predicted to be required for growth [44] and to localise to the plasma membrane [39]. Here, we show that 3'UTR mediated regulation of ZFT is important for parasite replication under low iron conditions, supporting the hypothesis that this gene is involved in iron transport. Interestingly, the stabilisation provided by the native UTR was transient, with levels maintained at 18 h but ZFT levels falling significantly at 24 h, in contrast to the tdTomato reporter. This may reflect differences in the promotor activity, but more likely represents differences in protein stability, suggesting that further layers of regulation may be regulating ZFT function. This has previously been observed for human ZIP14 which has a high endogenous rate of turnover which is increased in the absence of iron [76]. Nonetheless, this transient increase in protein levels was sufficient to confer increased growth in low iron, highlighting the potential importance of regulation. Future work will seek to functionalise this transporter and to expand on this regulation.

In summary, we have found that an IRE-like sequence in the 3' UTR of a proposed metal transporter confers iron responsivity. This responsivity is specific to iron depletion, and occurs first by stabilising the mRNA transcript, leading to accumulation of the protein. Iron-responsivity is modulated by an 18 nucleotide stretch within the predicted IRE. We find that although there is evidence of aconitase binding to transcripts in *T. gondii*, it is likely not regulating the stability of *zft*. However, regulation of *zft* by the native UTR is important for survival under low iron conditions. These data provide some of the first validation of iron-mediated regulation in *Toxoplasma gondii* and its importance to the parasite.

## Methods

### *T. gondii* and host cell maintenance

*Toxoplasma gondii* tachyzoites were grown in human foreskin fibroblasts (HFFs) cultured in Dulbecco's modified Eagle's medium (DMEM) maintained at 37 °C with 5% $CO_2$ that was supplemented with either 3% (D3) or 10% (D10) heat-inactivated foetal bovine serum (FBS), 2 mM L-glutamine and Penicillin-Streptomycin (50 U/ml).

### Construction of *Toxoplasma gondii* lines

The tdTomato$_{dhfr}$, tdTomato$_{zft}$ and tdTomato$_{zftDIRE}$ reporter lines were generated by transfection [77] of 50 mg of plasmid containing Cas9 and sgRNA (primer 6) for the *uprt* locus as previously described [4] along with 100 µL of PCR product containing the relevant reporter gene cassette into the constitutively expressing RHDku80::mNeonGreen line [4]. tdTomato$_{zft-DIRE}$ was made using QuikChange II XL (Agilent) following manufactures instructions using primers 15 and 16. After transfection, parasites were cultured in D3 media containing 5 mM FUDR for 5 days to select for disruption of the *uprt* locus as described previously [78]. Following selection, tdTomato parasites were collected by FACS using an Aria III (BD Biosciences), sorted directly into a 96 well plate pre-seeded with HFFs and incubated at 37 °C with 5% CO2 for 5–7 days. Positive fluorescent plaques were then screened by PCR for the disruption of the *uprt* locus and the correct insertion of the reporter cassette (primers 13 and 14). Fluorescent

parasites were resorted as above to ensure purity of populations. For all sgRNA sequences and primers see S13 Table.

For the generation of the ZFT-3HA$_{zft}$ strain, the 3HA epitope sequences, ZFT 3' UTR and chloramphenicol resistance gene was amplified by PCR from plasmid LIC-3HA$_{zft}$-CAT (modified plasmid from Sheiner et al., 2011[79]) to generate the repair template (using primers 22 and 23) which includes the upstream homologous region of ZFT and downstream homologous regions around the stop codon. F3ΔHX parasites were then transfected using a Gene Pulse BioRad electroporator with 30 μg of the purified repair template PCR product and 40 μg of a plasmid containing Cas9 guide (primer 26) targeting the 3' end of the genomic sequence of ZFT (S13 Table). Transgenic parasites were selected with chloramphenicol, cloned by limiting serial dilution. Clones were validated by PCR using primers 24 and 25.

## Chemicals

*N,N,N',N'*-tetrakis-(2-pyridylmethyl)ethylenediamine (TPEN) (Merck, 616394), made up in DMSO and used at 5 μM. Tetrathiomolybdate (TTM), made up in PBS and used at 25 μM (Merck, 323446). 2,2'-bipyridine (BIP) (VWR, SIALD216305), used at 100 μM, pyridoxal isonicotinoyl hydrazone (PIH) (Abcam, ab145871) used at 100 μM, Deferoxamine mesylate salt (Sigma, D9533), concentration as indicated in the text.

## Transcriptomics analysis using bulk RNA-sequencing

T75 flasks of HFF were incubated in either standard D3 or D3 supplemented 100 μM DFO for 24 h prior to infection. Each T75 was then infected with $6-7 \times 10^6$ cells of RHΔKu80 parasites and cultured for 24 hours prior to parasite collection. Parasites were pelleted by centrifugation at $1500 \times g$ for 10 min, washed and pellets stored at −80 °C until required. RNA was extracted from the pellets using the RNAeasy kit (Qiagen) according to the manufacturer's instructions. RNA libraries were then prepared using Illumina Stranded mRNA library preparation method and sequenced at $2 \times 75$ bp to an average of more than 5 million reads per sample. Raw sequencing data (FASTQ format) was processed using the Galaxy public server hosted by EuPathDB (https://veupathdb.globusgenomics.org/). FastQC and Trimmomatic were used for quality control and to remove low quality reads (where Q < 20 across 4 bp sliding windows) and adapter sequences [80]. The filtered reads were aligned to the *T. gondii* ME49 genome v64 using HISAT2 [81]. These sequence alignments were used to identify reads uniquely mapped to annotated genes using Htseq-count. Differential expression analysis was performed in R using DESeq2 [82]. Raw FASTQ files available from the EBI ENA online server under project PRJEB67890 and PRJEB83013.

## Identification of iron response element-like sequences in the *T. gondii* transcriptome

The *Toxoplasma gondii* reference ME49 transcriptome (v51, ToxoDB.org) was used to identify IRE sequences [83]. At the time of analysis many *T. gondii* transcripts lack annotated UTRs, for these genes a 1kb region up- and down-stream of the transcription start site [84]/ stop codons was included in the bioinformatic search. A 1 kb region was selected based on the average annotated UTR size which we found to be approximately 750 bp as previously reported [35]. These sequences were then entered into the SIREs online web tool (http://ccbg.imppc.org/sires/), accessed Jan 2023 [12]. Only sequences with a 'high' or 'medium' score were selected for further investigation.

## Flow cytometry

Reporter parasites were grown in untreated HFF cells or HFF cells pretreated for 24 h with 100 µM DFO (unless indicated) for 24 h (unless indicated). Cells were washed, parasites mechanically released from host cells and filtered through a 3 mm filter to remove host cell debris. Parasites were pelleted by centrifugation and resuspended in Ringers solution (115 mM NaCl, 3 mM KCl, 2 mM $CaCl_2$, 1 mM $MgCl_2$, 3 mM $NaH_2PO_4$, 10 mM HEPES, 10 mM glucose), then analysed on a BD Celesta analyser using FACSDiva software (BD Biosciences). Data acquired from at least 30,000 parasites was gated on forward and side scatter and on green fluorescence. All data were analysed using FlowJo v10 (BD Biosciences) and the geometric mean reported.

## Reporter gene expression by microscopy

Confluent HFF cells grown on coverslips were untreated or pretreated for 24 h with 100 µM DFO prior to infection with reporter cell lines. Infected cells were incubated for 24 h before fixation with 4% paraformaldehyde at room temperature for 20 mins. Cells were blocked and permeabilised in blocking buffer (2% bovine serum albumin, 0.05% Triton X-100 in PBS) for 20 mins and mounted onto slides using Fluoromount with DAPI (Southern Biotech). Micrograph images were obtained using a DeltaVision widefield (Applied Precision) or Leica DiM8 (Leica Microsystems) microscope and processed using SoftWoRx and Fiji [85]. Vacuolar tdTomato fluorescence was quantified using a custom ImageJ macro and presented as a ratio compared to mNeonGreen expression.

## NanoLuciferase assay

Confluent HFF cells grown were untreated or treated with 0.5 - 100 µM DFO for 24 h prior to infection. Each dish was then infected with $6–7 \times 10^6$ cells of nanoLuc$_{zft}$ reporter parasites or RHΔKu80:mNeonGreen parasites and cultured for 24 h prior to parasite collection. Parasites were mechanically released from host cells and filtered through a 3 mm filter to remove host cell debris. For each condition, $1 \times 10^7$ parasites per well were added to a white-bottom 96-well plate in triplicate. Uninfected HFFs were prepared in the same way as the parasites and used as background fluorescence/luminescence control. Fluorescence was measured using PHERAstar FS microplate reader (BMG LABTECH). Subsequently, nanoluciferase assay was performed using Nano-Glo Luciferase Assay System (Promega) according to the manufacturer's instructions. Briefly, Nano-Glo Luciferase Assay Substrate was diluted 1:50 with Nano-Glo Luciferase Assay Buffer, added to the plate at 50 µl/well and incubated for at least 5 min before luminescence measuring. The luminescence levels (in lumens; LUM) were measured using PHERAstar FS microplate reader (BMG LABTECH).

## Alkaline stress reporter assay

Parasites were seeded onto HFFs and allowed to invade for 4 hours, following this media was replaced with either R3 media (RPMI with 3% FBS, 2 mM L-glutamine and Penicillin-Streptomycin (50 U/ml)) or conversion media (RPMI with 1% FBS, 2 mM L-glutamine and Penicillin-Streptomycin (50 U/ml), 50 mM HEPES pH 8.2). Parasites were incubated for 24 h or 48 h before collection and prepared for flow cytometry or fixation followed by immunofluorescence staining for BAG1, as described above.

## Detection of BAG1 by immunofluorescence assay

Confluent HFF cells grown on coverslips were untreated or pretreated for 24 h with 100 µM DFO prior to infection with reporter cell lines. Infected cells were incubated for 24 h before

fixation with 4% paraformaldehyde at room temperature for 20 mins. Cells were blocked and permeabilised in blocking buffer (2% bovine serum albumin, 0.05% Triton X-100 in PBS) for 1 hour before incubation with rabbit anti-BAG1 antibody ([86,87] generous gift from Lilach Sheiner) diluted 1:1000 in blocking buffer for 1 hour. Coverslips were then washed three times in PBST (0.05% Triton X-100 in PBS) before incubation with goat anti-rabbit Alexa647 secondary antibody diluted 1:1000 in blocking buffer (ThermoFisher) for 1 hour. Finally, coverslips were washed again in PBST 3 times before being mounted onto slides using Fluoromount with DAPI (Southern Biotech) and imaged on the Zeiss LSM 880 Confocal microscope with Zeiss Zen 2 black software. Images were processed using Fiji [85].

## Reporter gene expression by RTqPCR

RT-qPCR was used to assay relative *zft* and *tdTomato* expression. HFF cells were untreated or treated with DFO (100 μM) for 24 hours prior to infection with the reporter lines (RHΔKu80, ΔUPRT:tdTomato$_{dhfr}$, ΔUPRT:tdTomato$_{zft}$ or ΔUPRT:tdTomato$_{zftΔIRE}$). Parasites were cultured as indicated before mechanical lysis of host cells and filtration through a 3 mm filter to remove host cell debris. Parasites were then pelleted by centrifugation at 1500 x g for 10 min, the media removed and cell pellets stored at −80 °C. Total RNA was extracted from parasite pellets using the RNAeasy Mini kit (Qiagen) and DNAse I 1 U/μL (Invitrogen) treated for 15 mins at room temperature followed by DNAse denaturation at 65 °C for 20 mins. cDNA synthesis was performed with High-Capacity cDNA Reverse Transcription Kit (Applied Biosystems) according to manufacturer's instructions using 1 mg of DNAse I treated RNA. RT-qPCR was carried out on the Applied Biosystems 7500 Real Time PCR system using Power SYBRgreen PCR master mix (Invitrogen), 2 ng of 1:10 diluted cDNAs per reaction and the following cycling conditions: 95 °C for 10 minutes, 40 cycles of 95 °C for 15 seconds, 60 °C for 1 minute. RT-PCR primers sequences can be found in S13 Table (primers 7, 8, 9, 10, 11, 12, 18 and 19). Reactions were run in triplicate from three independent biological experiments. Relative fold changes for treated vs. untreated cells was calculated using the Pfaffl method [88] using *actin* as a housekeeping gene control. Graphpad Prism 9 was used to perform statistical analysis.

## Iron chelator proliferation assay

Iron chelators (DFO, PIH and BIP) were tested for parasite inhibition as previously described [4]. Briefly, confluent HFF monolayers grown in clear bottomed 96 well plates were infected with 5000 ΔKu80:tdTomato [4] parasites/well. Invasion was allowed to occur for 2 h before the media was removed, and chelators added and serially diluted. Cells were incubated for 4 days before parasite proliferation quantified by PheraStar plate reader. For non-fluorescent parasites, confluent HFFs grown in clear 96 well plates were infected and treated as above, fixed with ice cold methanol for 10 mins at room temperature and stained with crystal violet (12.5 g crystal violet in 125 ml ethanol, diluted in 500 ml 1% ammonium oxalate) for 1 h and washed with water. Host monolayer disruption was then quantified by reading absorbance at 590nm using by a PheraStar plate reader. Results were normalised to untreated or uninfected wells and inhibition curves plotted by non-linear regression using Prism (v10).

## Aconitase activity assay

Aconitase activity was quantified based the production of NADPH from *T. gondii* [66]. Untreated parasites or those treated for 24 h with DFO were harvested as described above, pelleted by centrifugation and resuspended in aconitase activity buffer (50 mM Tris–HCl, pH 7.4, 6 mM sodium citrate, 0.2% Triton X-100, 0.6 mM MnCl$_2$) before lysis by three rounds of

freeze-thaw. Protein content of lysates was quantified by Bradford assay (ThermoScientific, B6916) following manufacturer's instructions. 10 mg of parasite lysate was combined with 0.2 mM NADP and 0.004 U of isocitrate dehydrogenase (Sigma, I2002) in a total volume of 100 μl. In each assay, 10 μl of disrupted and filtered host cell debris was included to ensure no contamination by host cell enzymes present. The absorbance was read at 340 nm at 25 °C every 5 mins for 1 h on a PheraStar plate reader. After normalization to blanks, the slope was calculated from the linear section of the curve and protein activity (c) calculated using c = $\text{slope}_{final}/\epsilon \cdot$ b with $\epsilon$ of NADP = 6.22 mM$^{-1}$ cm$^{-1}$ and path length (b) set to 1. Results represent four independent experiments performed in quadruplicate.

## Immunoprecipitation of *T. gondii* proteins and interacting RNAs

Parasites (RHΔku80:mNeonGreen-Ty or RH,ACN-Ty) were cultured for 18 hours in HFFs which had been pre-treated in D3 with 100 μM DFO for 24 hours. To collect the parasites host cell monolayers were scraped, passed through a syringe with 23G gauge needle three times, filtered through a 3 mm polycarbonate filter membranes to remove host debris and then pelleted by centrifugation. Cell pellets were frozen and stored at −80 °C prior to further processing. For immunoprecipitation, 5 x 10$^8$ parasites were lysed for 30 minutes on ice in 500 mL lysis buffer (50 mM Tris pH7.4, 150 mM NaCl, 1 mM EDTA, 1 mM EGTA, 1% Triton-X100 with Halt protease inhibitor (ThermoFisher, 78429) and SUPERase RNAse inhibitors 1 U/mL). Lysates were centrifuged at 13,000 rpm at 4 °C for 15 mins and the supernatants (cleared lysate) blocked for 1 h at 4 °C with 50 mL of Pierce Protein-G magnetic beads with end-over-end rotation. Lysates were then incubated for 3 h at 4 °C with 50 mL of Pierce Protein-G beads bound with mouse anti-Ty antibody (Invitrogen Ty1 Tag Monoclonal BB2). Beads were washed 6 times with wash buffer (200 mM Tris pH 9, 100 mM Potassium acetate, 0.5% (v/v) Triton x-100, 1 mM EGTA, 0.1% Tween 20, 2.5 mM Dithiothreitol). Total RNA was then extracted from the washed beads by Trizol-chloroform extraction, following manufacturer's instructions. Extracted RNA was treated with DNAse I (Invitrogen) as above before proceeding to cDNA synthesis. For qPCR: cDNAs were produced using High-Capacity cDNA Reverse Transcription Kit (Applied Biosystems), according to the manufacturer's instructions. Resulting cDNAs were diluted 1:5 prior to qPCR which was performed as described above. For primer sequences see S13 Table (primers 7, 8, 18, 19, 20 and 21). For RIP-seq: The cDNA was produced from RNA as follows. First strand synthesis: SuperScript IV reverse transcriptase (Thermo) and random hexamer primers. Resulting DNA/RNA were purified using Beckman Coulter AMPure XP beads (RNAse free) using manufacturer's instructions. Second strand synthesis: DNA/RNA hybrids were digested with RNAse H (NEB) and the second strand synthesized using DNA Pol I (NEB) in NEB buffer 2. Resulting cDNAs were fragmented by sonication on ice (9x 30 seconds with 60 second gap between pulses, Hologic Diagenode Biorupterä). Sequencing libraries were prepared using the NEBnext library prep kit according to manufacturer's instructions. The resulting libraries were sequenced using a P3 flow cell on a NextSeq2000 sequencer. Samples were sequenced, paired end (2x100bp), to a depth of at least 5 million reads per sample. Downstream QC and analysis of the resulting reads was performed as described above for in the 'Transcriptomics analysis using bulk RNA-sequencing' methods section.

## mRNA stability assay

HFF cells were pre-treated for 24 hours as above before being infected with tdTomato$_{dhfr}$ or tdTomato$_{zft}$ reporter parasites. At 18 h post infection, 10 mg/ml Actinomycin D was added to the culture media in the treatment dishes. Parasites were collected at 0, 1, 3 and 5 hours

post-Actinomycin D treatment. At each collection, host cells were mechanically lysed and filtered through a 3 mm membrane to remove debris. Parasites were pelleted by centrifugation and immediately frozen on dry ice for storage at −80 °C. RNA was extracted using a Qiagen RNAeasy kit according to manufacturer's instructions. RNAs were then processed as described above for RTqPCR or sequencing. All primer sequences can be found in S13 Table (primers 9, 10, 18 and 19). All data was plotted in Prism (v10) and non-linear regression (one phase decay) performed to determine mRNA half-life. The same infection and collection protocols were used for the transcriptome-wide stability assay. RNA libraries were then prepared using Illumina Stranded mRNA library preparation method and sequenced at 2 x 100 bp to an average of more than 10 million reads per sample. Raw sequencing data (FASTQ format, available on the ENA through accession PRJEB83011) was processed using the Galaxy public server hosted by EuPathDB (https://veupathdb.globusgenomics.org/). FastQC and Trimmomatic were used for quality control and to remove low quality reads (where Q < 20 across 4 bp sliding windows) and adapter sequences [80]. The filtered reads were aligned to the *T. gondii* ME49 genome v64 using HISAT2 [81]. These sequence alignments were used to identify reads uniquely mapped to annotated genes using Htseq-count. To be included in downstream analysis transcripts must have at least 1 read in all samples. Counts were normalised and differential abundance analysis was performed in R using DESeq2 using normalized counts [82]. To be included in downstream analysis transcripts must have decayed to <75% of the abundance at t = 0 hours in the parasites cultured in standard iron conditions, this threshold was computed as 1 standard deviation of the average decay of all transcripts. To aid data interpretation, we binned transcripts meeting this threshold based on their decay kinetics. Bin 1 contains transcripts which decayed to 50% of t = 0 by 3 h and then decay further. Bin 2 contains transcripts which decayed to 50% of t = 0 by 3 h but did not decay further. Bin 3 contains transcripts which did not decay by 50% in the first 3 h. Finally, we determined which transcripts showed statistically significant differential abundance relative to the t = 0 timepoint, for each condition using Wald tests with Bonferroni correction.

## Motif discovery and enrichment

Whole transcript sequences of interest from the TGME49v64 genome version [83] were in inputted into XSTREME [55,56] as the 'test set'. The entire TGME49v64 transcriptome was provided as the 'control set'. Motif discovery and enrichment was aided by RNA motif data from eukaryotes [89] as provided by XSTREME. The tool scanned for motifs of length between 6 and 30 bases (the maximum enabled by the tool) and stopped searching when motif scores breached the e-value <0.05 significance limit. All other settings were kept as the defaults.

## Plaque assay

500 parasites of the indicated strains were allowed to infect confluent monolayers of HFF cells. After 7 days of undisturbed replication, monolayers were washed with PBS, fixed with ice cold 70% ethanol and stained using crystal violet stain (as above) for 1 h at room temperature before washing with distilled $H_2O$ and imagining. Plaque area was quantified using ImageJ.

## Detection of ZFT protein by western blot

Parasites were filtered and collected by centrifugation at 2000 x g for 10 minutes and then lysed with RIPA lysis buffer (150 mM sodium chloride, 1% Triton X-100, 0.5% sodium deoxycholate, 0.1% sodium dodecyl sulfate (SDS) and 50 mM Tris, pH 8.0) on ice for 30 minutes. Lysates were resuspended with 5x Laemmli buffer (10% SDS, 50% glycerol, 300mM TrisHCL

pH 6.8 and 0.05% bromophenol blue) and boiled at 95 °C for 5 minutes. Samples were then separated on a 10% SDS-PAGE gel for 20 minutes at 120 V then 175 V for 1 hour. Proteins were then wet transferred to nitrocellulose membrane in Towbin buffer (0.025 M Tris, 0.192 M Glycine, 10% methanol) for 60 minutes at 250 mA and blocked at room temperature in 5% milk in 0.1% Tween/PBS. Blots were then stained with primary antibodies overnight at 4 °C (rat anti-HA (Merk 11867423001) at 1:500 and guinea pig anti-CDPK1 at 1:10,000 [35]), followed by secondary fluorescent antibodies at room temperature for 1 hour (anti-rat HRP 1:5000, Abcam, ab6734 and 1:10,000 and goat anti-guinea pig coupled to IRDye 680 1:10,000). PageRuler Prestained Protein Ladder (ThermoScientific) was used as a molecular weight marker. ZFT-HA was detected using Pierce ECL Western Blotting Substrate (Thermo Scientific) and imaged using the Invitrogen iBrightFL1000 Imaging System. CDPK1 was detected using the LI-COR Odyssey LCX system.

## Supporting information

**S1 Fig. A. Schematic of significantly changed GO-terms from RNAseq experiment comparing the transcriptomes of parasites cultured in low iron conditions.** Circle size represents the proportion of genes from that GO set which were enriched in the differentially expressed genes from our data set (gene ratio). The intensity of the circles represents the significance adjusted p-value (Bonferroni corrected) of the gene ratios. This analysis was performed with ToxoDB [83]. **B.** Volcano plot from RNAseq data comparing RHΔBFD1 cultured in 100 μM DFO for 24 hours to standard conditions. Adjusted p-values from the Wald test with Benjamini and Hochberg correction. Cut-offs shown with dashed lines are p-adj < 0.05 and $\log_2$ fold change of >2 or <−2. **C.** Correlation plot showing that response of parental and ΔBFD1 to DFO treatment is highly correlated (Spearman's correlation, $R^2$ = 0.89, p < 0.0001). (TIF)

**S2 Fig. PCR confirmation of iron reporter lines. A.** Alignment showing conservation of IRE sequence in the 3'UTR of *zft* across *T. gondii* strains. Alignments were performed using T-Coffee online alignment tool M-Coffee [90]. **B.** Schematic of the cloning strategy for adding the reporter cassettes into the *uprt* locus using CRISPR-Cas9 **C-E.** PCRs showing successful amplification of the full reporter tdTomato$_{dhfr}$ (**C**, expected size 4.3kb), tdTomato$_{zft}$ (**D**, expected size 4.3kb), tdTomato$_{zftΔIRE}$ (**D**, expected size 4.3kb), and tdTomato$_{isu1}$ (**E**, expected size 3.2kb), cassettes into the *uprt* locus of RHΔKu80:mNeonGreen parasites. Parental line included as negative control **F.** Overlapping histogram showing tdTomato fluorescence in untreated tdTomato$_{dhfr}$ (black), tdTomato$_{isu1}$ (blue) and tdTomato$_{zft}$ (red) reporter lines as measured by flow cytometry. **G.** PCR showing successful amplification of the full reporter nanoLuc$_{zft}$ (expected size 4.1 kb). **H.** Luminescence experiment showing nanoLuc$_{zft}$ expresses detectable luciferase under basal conditions. **I.** Overlapping histogram showing tdTomato fluorescence in untreated tdTomato$_{zft}$ (red) and tdTomato$_{zftΔIRE}$ (dark red) reporter lines as measured by flow cytometry. **J.** IFA images showing expression of bradyzoite marker BAG1 at 24 h post alkaline stress. Scale bar 5 μm. (TIF)

**S3 Fig. Aconitase associates with *zft* transcripts. A.** PCA plot showing clustering of biological replicates from mRNA stability assay. **B.** Schematic of tagging scheme in ΔKu80:mNeonGreen-Ty and ACN-Ty parasites. **C.** Immunofluorescence of ΔKu80:mNeonGreen-Ty and RHΔKu80, ACN-Ty parasites grown in standard culture or 100 μM DFO for 24 hours. Anti-TOM40 included as a mitochondrial marker and DAPI as DNA marker. Scale bars 5 μm. **D.** Aconitase activity assay of untreated or DFO treated parasites. Results from 4 independent experiments, ± SD. Lysed and filtered host cells are included to demonstrate minimal carryover

of host aconitase activity. *p* value from paired t test. **E.** Normalised counts for *zft* and *rng2* transcripts from RNAseq dataset comparing parasites after 24 hours treatment with 100 μM DFO, compared to untreated parasites. Points represent 3 independent experiments, bars at mean ± SD. Two (**F** and **G**) further biological replicates showing ACN-Ty interacts with *zft* mRNA. Blots showing the immunoprecipitation of mNeonGreen-Ty and aconitase-Ty from *T. gondii* parasites. Wash – the output of the 6th and final washing step. CDPK1 included as cytosolic control. From the corresponding pull down, DNA-agarose gel showing qPCR products amplified from reverse transcribed RNA from either lysates or RNA co-precipitated with mNeonGreen-Ty and ACN-Ty proteins.
(TIF)

**S4 Fig. Aconitase is highly conserved between apicomplexans and mammals. A.** Amino acid sequence alignment of the *Toxoplasma gondii* aconitase hydratase ACN/IRP (TGME49_226730), *Plasmodium falciparum* aconitase hydratase (PF3D7_1342100) and human IRP1 (UniProt: P21399) and IRP2 (UniProt: P48200) in ClustalW format made using T-Coffee [90]. Green triangles indicate the residues interacting with the FeS cluster. Purple triangles indicate residues shown to interact with ferritin mRNAs in human IRP1 [61]. **B.** Alphafold [91,92] structure prediction showing high structural conservation between TgACN (magenta) and HsIRP1 (pdb 2B3X) (cyan), including around the FeS coordination core (yellow, inset).
(TIF)

**S5 Fig. Plaque area indicates no growth defect of ZFT-HA*sag1* cells. A.** Schematic of tagging scheme of ZFT-HA parasites. **B.** PCR confirmation of tagged lines. Expected size as indicated in (**A**). **C.** Area of plaque of four independent experiments. Each point is a plaque, line at mean, ± SD. Red dots indicate average plaque size for each experiment. *p* > 0.05 from one way ANOVA.
(TIF)

**S1 Table. RNAseq results from experiment comparing parasites cultured for 24 hours in 100μM DFO to untreated parasites.**
(XLSX)

**S2 Table. Genes predicted by MetalPredator to encode proteins with iron sulphur clusters.** Log$_2$ fold changes and adjusted *p* values from RNAseq experiment comparing parasites cultured for 24 hours in 100 μM DFO to untreated parasites.
(XLSX)

**S3 Table. FeS synthesis pathway genes [2] with log$_2$ fold changes and adjusted *p* values from RNAseq data.**
(XLSX)

**S4 Table. Genes upregulated in *T. gondii* bradyzoites (log$_2$ fold changes > 2, adjusted *p* value < 0.05) from Waldman *et al.* 2020 in low iron conditions.**
(XLSX)

**S5 Table. RNAseq data from ΔBFD1 parasites, untreated vs. 100 μM DFO for 24 hours.**
(XLSX)

**S6 Table. Raw output file from SIREs web server [12] identifying putative IRE-like sequences in curated *T. gondii* transcriptome.**
(XLSX)

**S7 Table. Curated list of genes containing putative IRE-like sequences in the *T. gondii* transcriptome containing only high quality and medium quality scoring sequences as identified using SIREs [12].**
(XLSX)

**S8 Table. Curated list of genes containing putative IRE-like sequences in the *T. gondii* transcriptome containing only high quality and medium quality scoring sequences, limited to those found within untranslated regions [12].**
(XLSX)

**S9 Table. Curated list of genes containing putative IRE-like sequences in the *T. gondii* transcriptome containing only high quality and medium quality scoring sequences with from motif classes I, II and VIII, found within the untranslated regions - as identified using SIREs.**
(XLSX)

**S10 Table. RTqPCR Ct values from RIP-qPCR experiments to look for transcripts associated with *Toxoplasma gondii* aconitase.**
(XLSX)

**S11 Table. Transcripts significantly enriched in Aconitase-Ty IP and not mNeonGreen-Ty IP and details of discovered enriched motifs.**
(XLSX)

**S12 Table. Transcripts which demonstrated differential stability between the untreated and iron deprived conditions and details of on discovered enriched motifs.**
(XLSX)

**S13 Table. Oligonucleotides used in this study.**
(XLSX)

## Acknowledgments

We would like to thank Aarti Krishnan and Dominique Soldati for the iACN:Ty line, Sebastian Lourido for the ΔBFD1 parasite line and Lilach Sheiner for the anti-TOM40 and anti-BAG1 antibodies. Additionally, we would like to thank Shikha Shikha for assistance with AlphaFold and Katarzyna Modrzynska for her guidance and RIP-qPCR protocol. We would like to acknowledge www.ToxoDB.org, without which this work would not have been possible.

## Author contributions

**Conceptualization:** Megan A. Sloan, Clare R. Harding.

**Data curation:** Adam Scott, Clare R. Harding.

**Funding acquisition:** Megan A. Sloan, Clare R. Harding.

**Investigation:** Megan A. Sloan, Adam Scott, Dana Aghabi, Lucia Mrvova, Clare R. Harding.

**Methodology:** Megan A. Sloan, Adam Scott, Clare R. Harding.

**Supervision:** Clare R. Harding.

**Writing – original draft:** Megan A. Sloan, Clare R. Harding.

**Writing – review & editing:** Megan A. Sloan, Dana Aghabi, Lucia Mrvova, Clare R. Harding.

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
