## [Decision Letter · Decision Letter 0]

24 Feb 2024

Dear Dr. Harding,

Thank you very much for submitting your manuscript "Keeping FIT: Iron-mediated post-transcriptional regulation in Toxoplasma gondii" for consideration at PLOS Pathogens. As with all papers reviewed by the journal, your manuscript was reviewed by members of the editorial board and by several independent reviewers. In light of the reviews (below this email), we would like to invite the resubmission of a significantly-revised version that takes into account the reviewers' comments.

The reviewers concur that the study is addressing an important topic and is well-presented. They however raised some issues that are necessary to be addressed for acceptance and others that are optional but would be nice to have. To bring the study to a more satisfying conclusion:

- Validate the reporter findings by mutating the endogenous 3’UTR of FIT and showing that its protein levels are no longer affected during iron depletion. - Test the IRE positioning effect on its function. - Demonstrate that ACN is binding to the IRE transcripts by performing either more stringent IPs or a high throughput sequencing of the transcripts pull-down in the IP. - Address the concerns about statistical analyses

We cannot make any decision about publication until we have seen the revised manuscript and your response to the reviewers' comments. Your revised manuscript is also likely to be sent to reviewers for further evaluation.

Sincerely,

Dominique Soldati-Favre

Section Editor

PLOS Pathogens

Michael Malim

Editor-in-Chief

PLOS Pathogens

orcid.org/0000-0002-7699-2064

Reviewer's Responses to Questions

**Part I - Summary**

Reviewer #1: This manuscript addresses mechanisms by which the parasite Toxoplasma gondii responds to depleted iron levels. Results show that Toxoplasma experiences changes in gene expression when iron is scare and uses the post-transcriptional IRE/IRP system, which is well-characterized in a number of other eukaryotes and has been described for Plasmodium. The authors complete an impressive suite of well-controlled experiments to show conservation of this regulatory pathway in Toxoplasma.

Reviewer #2: This is a manuscript from Sloan et al. entitled "Keeping FIT: Iron-mediated post-transcriptional regulation in Toxoplasma gondii". It describes the identification and characterization of an Iron Response Element (IRE) in the 3'UTR of the TGME49_261720 transcript using a reporter system. Regulation of gene expression by Iron has been shown recently by the same group. However, the molecular mechanisms underlying this regulation have not been explored so far. Although IREs are widespread in eukaryotes and their sequence is relatively conserved, their functionality in the parasite was not investigated. This work therefore explores a regulation mechanism that seems common to eukaryotes. The authors provide convincing data on the functionality of the IRE found in the 3'UTR of the TGME49_261720 transcript in a heterologous reporter system. However, only one of the 142 IREs that were found in transcripts is functional (out of 2 tested). Moreover, the authors fell short of confirming the role of Aconitase in this regulation, a key conserved mechanism in Iron response.

Reviewer #3: this is an interesting paper about a topic that has not been well explored. data are well presented and organized. I only have minor comments. Nice work.

**Part II – Major Issues: Key Experiments Required for Acceptance**

Reviewer #1: Major concern:

1. There is no evidence that FIT protein levels increase in an IRE-dependent way. Endogenous FIT should be epitope-tagged using its native 3’UTR or its 3’UTR deleted of its IRE sequence. This experiment seems necessary to verify the conclusions drawn from the reporter assays and validate the model proposed by the authors.

Suggested experiments to improve study and strengthen conclusions:

1. Transcriptomics data suggest that iron deprivation induces bradyzoite differentiation (Fig. 1). Authors should consider a simple differentiation assay to confirm signature features of tissue cyst formation (and at what frequency) during iron deficiency. This is important not only to bolster the author’s claim but also to assess whether iron depletion may be a better inducer of in vitro bradyzoite development than current methods.

2. Chelation of other metals serves as a nice control to iron deprivation (Fig 3); it may be of interest to also test if a different inducer of bradyzoite development activates the FIT reporter as a further control on its specificity.

3. The idea that IRE position effects its function is an interesting hypothesis; authors may want to consider designing a reporter to formally test this (i.e. move the IRE-like sequence of iscu closer to the stop codon). Alternatively (or in addition), authors could consider a reporter that moves the IRE sequence of FIT further downstream in the 3’UTR.

4. An independent line of evidence would strengthen the conclusion that ACN binds to FIT at the IRE. For example, immobilizing the TGME49_261720 3’UTR with and without the IRE and passing lysate from ACN-Ty expressing parasites over it.

Reviewer #2: Major concerns

The fit transcript is downregulated in the absence of iron. How do the authors reconcile their data on the functionality of this IRE with this measured expression in the absence of Iron? How many of the 142 transcripts containing an IRE and regulated by Iron are downregulated?

The authors only characterized 2 different IREs and only one of them seems functional. It would be better to expand the number of IRE tested to confirm their functionality in this parasite.

Some critical information is missing when exploring the IRE functionality. The distance to the STOP codon could be explored as well as a much-detailed mutagenesis analysis of the IRE. A key experiment is also missing: what happens to the reporter expression when the IRE is placed in the dhfr 3'UTR context? This would help in understanding the role of the motif itself in the observed regulation.

It is rather surprising that the authors choose to use tdTomato as a reporter. The long half-life of the protein and the reduced dynamic range of fluorescence intensity are not fitted to the measurement of transcript stability or translation efficiency. Luciferase is usually used for this purpose. It would be interesting to reiterate some of the experiments with a luciferase reporter to test whether a bias was introduced in the measurements.

In Figure 5B, the statistical test of the comparison between the tdTomatofit and the tdTomatofit∆IRE is not presented. It seems that the level of expression of the two constructs is not significantly different. If this is the case, this strongly attenuates the importance of the IRE in the response to iron stress.

The author failed to demonstrate that Aconitase is binding to the IRE transcripts since actin is also found in the transcripts of the IP fraction. In Figures 6E and 6F, although the rng2 (a cell cycle-dependent transcript) is absent from the IP, the presence of actin shows that the background is high in these experiments. Performing these experiments with more stringent washing steps may be helpful. It would be also more convincing to perform a high throughput sequencing of the transcripts pull-down in the IP to identify the variety of transcripts bound by Aconitase.

Reviewer #3: None

**Part III – Minor Issues: Editorial and Data Presentation Modifications**

Reviewer #1: Minor issues:

1. “fit” should be defined in abstract and in its first appearance in Introduction. FIT should be worked into the author summary as well since it appears in the title.

2. Many typos and errors in the text (missing punctuation, T. gondii not in italics in all places, extra words, “nt” instead of “not,” etc). Most references are numerals but at least one is a lead author’s name (Augusto 2021) and this paper is not listed in the references section.

3. Reword this incomplete sentence: “Selecting a predicted IRE from the 3’ UTR of a predicted, but unstudied iron transporter, TGME49_261720, which we name here FIT.”

4. The authors state in Results: “TGME49_261720 encodes an essential ZIP-domain containing protein (41) and is a putative plasma-membrane localised zinc/iron transporter (35). For these reasons, we renamed the gene FIT.”

Based on this statement, it isn’t clear how the authors derived the name “FIT” – please define exactly what “FIT” stands for; in addition, ZIP should be defined as “zinc/iron permease.”

5. This statement reads awkward: “As at the transcript level, there was no significant change in tdTomato fluorescence was observed in the tdTomatodhfr cells at any timepoint.”

6. A GO analysis of transcripts altered by iron depletion would provide a helpful view of which biological process may be affected by iron deprivation.

7. Fig 6 – “localisation was not significantly affected by iron depletion.” Where is this data shown?

8. Legend contains typos (e.g. CAN-Ty instead of ACN-Ty…damn autocorrect!)

9. It is mentioned that TGME49_261720 is conserved between Toxoplasma genomes; as the RH strain is used for experiments, it seems appropriate to include it in the alignment on Figure S1A.

10. Figure S1B-C: the expected size for each PCR product is not clear in the figure and legends.

11. Fig. 2D: In the manuscript, it is mentioned that there was no increase in the ratio of green:red fluorescence in the tdTomato-dhfr line, but there is p<0.0001 listed on the graph. Did the authors mean to indicate the change as “not significant”?

12. Figure 6G is cited as 6F in the main text and 6H as 6G.

13. Reference 36 does not appear to have been published in a peer-reviewed journal. Not sure of PLOS policy here.

14. It would be extremely helpful if authors adhered to a more standardized manuscript format that included more readable font (e.g. arial size 11 or 12), double spacing, and contiguous line numbers for easy referencing.

Reviewer #2: Minor concerns:

It is not clear which are the 142 transcripts with an IREs identified.

The bibliography is sometimes improperly formatted (for ex: Augusto, 2021)

Reviewer #3: "supporting a recent observation (36)," sentence fragment?

"This suggests that iron deprivation, similar to other nutrient deficiencies (38–40), can promote

Toxoplasma differentiation" there is a preprint on this exact topic that would be worth citing if Plos Pathogens allows that.

Iron depletion has different consequences on the growth and survival of Toxoplasma gondii strains

Eléa A. Renaud, Ambre J.M. Maupin, Yann Bordat, Arnault Graindorge, Laurence Berry, View ORCID ProfileSébastien Besteiro

doi: https://doi.org/10.1101/2023.12.21.572787

unsure what "loosely targeted effect of iron deprivation" means. I think authors should elaborate.

fig 2d has P<0.0001 but text says there was no signfiicant difference in the dhfr line. typo?

figure 6d,e,f: minor concern about statistical analyses here as I am not seeing anything about how these data were tested statistically or if they were at all.

methods: 3mm should be 3 micrometer? 1 mg should be 1 microgram?

PLOS authors have the option to publish the peer review history of their article (what does this mean? ). If published, this will include your full peer review and any attached files.

**Do you want your identity to be public for this peer review?** For information about this choice, including consent withdrawal, please see our Privacy Policy .

Reviewer #1: No

Reviewer #2: No

Reviewer #3: No
---

## [Decision Letter · Decision Letter 1]

21 Dec 2024

Dear Dr. Harding,

We are pleased to inform you that your manuscript 'Iron-mediated post-transcriptional regulation in Toxoplasma gondii' has been provisionally accepted for publication in PLOS Pathogens.

Best regards,

Dominique Soldati-Favre

Section Editor

PLOS Pathogens

Sumita Bhaduri-McIntosh

Editor-in-Chief

PLOS Pathogens

orcid.org/0000-0003-2946-9497

Michael Malim

Editor-in-Chief

PLOS Pathogens

orcid.org/0000-0002-7699-2064

Reviewer Comments (if any, and for reference):

Reviewer's Responses to Questions

**Part I - Summary**

Reviewer #2: The authors addressed most of my concerns. The manuscript is now acceptable for publication.

**Part II – Major Issues: Key Experiments Required for Acceptance**

Reviewer #2: Acces to the RNA-seq data should be provided.

**Part III – Minor Issues: Editorial and Data Presentation Modifications**

Reviewer #2: A better presentation of the result of the RIP-seq experiment will benefit the reader. As such they are difficult to distinguish from the RT-PCR experiment.

PLOS authors have the option to publish the peer review history of their article (what does this mean? ). If published, this will include your full peer review and any attached files.

**Do you want your identity to be public for this peer review?** For information about this choice, including consent withdrawal, please see our Privacy Policy .

Reviewer #2: No

---

## [Editor Report · Acceptance letter]

Dear Dr. Harding,

We are delighted to inform you that your manuscript, "Iron-mediated post-transcriptional regulation in Toxoplasma gondii," has been formally accepted for publication in PLOS Pathogens.

Best regards,

Sumita Bhaduri-McIntosh

Editor-in-Chief

PLOS Pathogens

orcid.org/0000-0003-2946-9497

Michael Malim

Editor-in-Chief

PLOS Pathogens

orcid.org/0000-0002-7699-2064